# Reviving Error Correction in Modern Deep Time-Series Forecasting

## Abstract

Modern deep-learning models have achieved remarkable success in time-series forecasting. Yet, their performance degrades in long-term prediction due to error accumulation in autoregressive inference, where predictions are recursively used as inputs. While classical error correction mechanisms (ECMs) have long been used in statistical methods, their applicability to deep learning models remains limited or ineffective. In this work, we revisit the error accumulation problem in deep time-series forecasting and investigate the role and necessity of ECMs in this new context. We propose a simple, architecture-agnostic error correction model that can be integrated with any existing forecaster without requiring retraining. By explicitly decomposing predictions into trend and seasonal components and training the corrector to adjust each separately, we introduce the Universal Error Corrector with Seasonal–Trend Decomposition (UEC-STD), which significantly improves correction accuracy and robustness across diverse backbones and datasets. Our findings provide a practical tool for enhancing forecasts while offering new insights into mitigating autoregressive errors in deep time-series models.

## 1 Introduction

Time-series forecasting is essential across numerous industries, including finance, healthcare, energy management, and supply chain optimization. In recent years, deep learning models have significantly improved the accuracy of time-series forecasting (Wu et al., 2023; Zeng et al., 2023; Wang et al., 2024a;b). They outperform traditional methods on real-world benchmarks by leveraging advanced feature extraction and data-driven representations (Siami-Namini & Namin, 2018; Qiu et al., 2024). Despite these advances, long-term forecasting remains a persistent challenge. One approach is to directly train the model to predict a fixed, large number of future steps in a single forward pass. However, this requires significantly larger models, often exhibits degraded accuracy, and is not scalable to arbitrary prediction lengths. A more flexible alternative is autoregressive inference, which generates future steps sequentially by conditioning on previously predicted values. Yet, this paradigm suffers from compounding errors, as inaccuracies introduced at earlier steps propagate and amplify over time (Moreno-Pino et al., 2023).

Error modeling has been studied in traditional time-series forecasting, with classical Error Correction Models (ECMs) addressing long-term relationships by using cointegration and making adjustments for deviations from equilibrium, defined as a stable long-run relationship that the system gradually returns to after short-term fluctuations (Hansen, 2003; Barigozzi et al., 2024). Similarly, classic methods like ARIMA, based on autoregressive processes, make forecasts by considering past observations, predictions, and errors (Makridakis & Hibon, 1997). However, classical ECMs differ fundamentally from the error correction needed in deep learning models. They adjust for deviations from equilibrium across multiple time series, making them difficult to apply directly to modern deep learning models, which require the correction of errors arising from internal processing and autoregressive prediction. While error correction has been explored for specific deep learning models in recent research, solutions often involve predefined error functions to refine predictions (Zhang et al., 2021) or the integration of error correction layers within forecasting pipelines (Liu et al., 2020; Li et al., 2024), necessitating costly joint training of both the correction module and the forecasting model. To our knowledge, *there exists no error correction model (ECM) that reliably improves a wide range of modern forecasters while treating the underlying forecasting backbone as a black-box*. The absence of such an ECM is potentially due to the already high performance of current forecast-

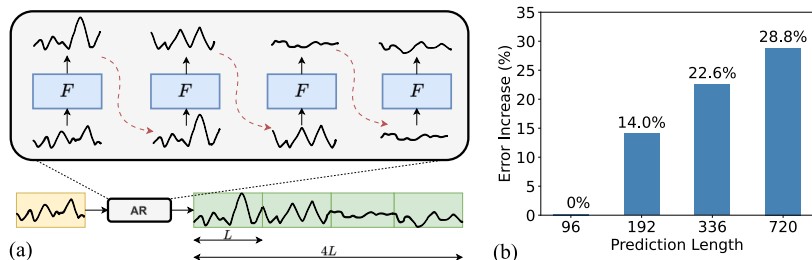

(a)                                        (b)

Figure 1: (a) Chunk-based autoregressive (AR) forecasting in time series. Given a forecaster $F$ with a fixed prediction window length $L$, which equals the input window size, the model's output must be recursively fed as input to predict a future horizon of length $4L$ (here, using $M = 4$ AR steps). (b) The relative increase in test prediction error when using model-predicted inputs instead of ground-truth, across 4 standard forecasting lengths: 96, 192, 336, and 720. Results are based on TimeMixer with $L = W = 96$ on the ETTh1 dataset.

ing methods, which makes ECMs redundant. Alternatively, it may stem from the risk of overfitting ECMs to specific model or dataset characteristics, thereby hindering their ability to perform well on test data (Nandutu et al., 2022). These considerations give rise to two key research questions under the autoregressive inference setting: (1) Are ECMs necessary for deep learning-based forecasting models? (2) How can ECMs be systematically integrated to generalize and improve the performance of state-of-the-art forecasting architectures?

In this paper, we study the feasibility of integrating ECM into deep forecasters. We propose the *Universal Error Corrector* (UEC), a simple framework that learns correction vectors from the inputs and outputs of pre-trained models. Once trained, UEC adjusts forecasts at inference to mitigate error accumulation over long horizons. While the UEC can be implemented as any machine learning model, we propose a specialized variant for time-series data, the *UEC with Seasonal-Trend Decomposition* (UEC-STD). Time-series forecasts often exhibit distinct long-term trends and short-term seasonal patterns, and the backbone forecaster may struggle differently with each. UEC-STD explicitly separates these components and learns targeted corrections for both, optimizing a weighted loss that balances trend and seasonal errors. The experimental results demonstrate that the UEC-STD consistently reduces error accumulation and significantly improves the accuracy of 3 deep forecasters with minimal additional computational cost. In summary, our contributions are: (i) We pioneer a universal error correction mechanism for modern forecasters without retraining the backbone; (ii) We design UEC-STD, a lightweight plug-in module that explicitly corrects trend and seasonal errors in time-series data; (iii) We validate UEC-STD across diverse datasets and models, showing consistent error reduction, efficiency, and insightful model analyses.

## 2 METHOD

To begin, we briefly introduce time-series forecasting. Here, the objective is to predict future values of a sequence based on historical observations. Let $\mathcal{D}_{train} = \{X_t\}_{t=1}^{T_{train}}$ represent the observed multivariate time-series data, where $X_t \in \mathbb{R}^D$ is the time-series values at time $t$, and $D$ is the number of variates. The forecasting task involves predicting future values over a horizon $L$ based on historical time-series observations. Specifically, let the past window of observations be represented as: $X_{t-W+1:t} = \{X_{t-W+1}, X_{t-W+2}, \ldots, X_t\}$ where $W$ is the look-back window length. Given this window, we aim to predict the future values of the time-series $X_{t+1}, X_{t+2}, \ldots, X_{t+L}$ using a model $F(\cdot)$: $\hat{X}_{t+1:t+L} = F(X_{t-W+1:t})$. The objective is to minimize the forecast error, often defined as the discrepancy between the predicted values $\hat{X}_{t+1:t+L}$ and the true future values $X_{t+1:t+L}$, by minimizing the forecasting loss functions such as MSE or Huber losses (Jadon et al., 2024).

### 2.1 CHUNK-BASED AUTOREGRESSIVE PREDICTION

Now, we formalize the autoregressive forecasting setup considered in this work. In this approach, during inference, when ground-truth data are unavailable for long-term forecasting, the model feeds

its previous predictions back as inputs (Shi et al., 2025). This can cause error propagation, as small prediction errors accumulate and amplify over time, leading to significant deviations.

Formerly, let $\hat{X}_t$ be the predicted value at time $t$, and $X_t$ the true value. In traditional autoregressive models, assuming we do not have the true data $X_t$, the process is: $\hat{X}_{t+1} = F(X_{t-W:t-1} \oplus \hat{X}_t)$ where $X_{t-W:t-1}$ is the history of observations up to time $t - 1$, $\hat{X}_t$ is the prediction for step $t$, and $\oplus$ is the concatenation of 2 time-series. In practice, we can apply a chunk-based autoregression that forecasts a window of $L$ time steps at a time (see Fig. 1 (a)). At the autoregression step $k = 0, 1, ..., M$, the predicted chunk $\hat{X}_{t+kL+1:t+(k+1)L}$ is fed back as input for the next prediction:

$$\hat{X}_{t+kL+1:t+(k+1)L} = \begin{cases} F(X_{t-W+1:t}) & \text{if } k = 0 \\ F(\hat{X}_{t+kL-W+1:t+kL}) & \text{if } k \geq 1 \end{cases} \tag{1}$$

Here, $M$ is the number of autoregressive steps needed to reach the desired horizon length $M \times L$. From now on, to simplify the notation, we set $\tau = t + kL$ as the chunk boundary at AR step $k$ starting from timestep $t$. Here, for any positive index $j$, if $\tau - W + 1 + j \leq t$:

$$\hat{X}_{\tau-W+1+j} = X_{\tau-W+1+j}. \tag{2}$$

By optionally using an overlapping window for the final step, chunk-based autoregression allows any model with a fixed prediction horizon $L$ to produce forecasts of arbitrary length $T$. For example, the last autoregressive step reads: $\hat{X}_{t+T-L+1:t+T} = F(\hat{X}_{t+T-L-W+1:t+T-L})$ where $M = \lceil \frac{T}{L} \rceil$ is the number of chunks and $T$ is the desired forecast length. For convenience, we denote the whole prediction using AR as:

$$\hat{X}_{t+1:t+T} = F_{AR}(X_{t-W:t-1}|T) \tag{3}$$

Despite its flexibility, this recursive formulation remains susceptible to error accumulation across chunks. As seen in Fig. 1 (b), the forecasting error grows with the number of autoregressive steps, compared to using ground-truth inputs at each step.

## 2.2 UNIVERSAL ERROR CORRECTION FRAMEWORK

**Autoregressive Correction Mechanism**  Let $\hat{X}_{t+1:t+L}$ represent the forecasted values, and let $\Delta\hat{X}_{t+1:t+L}$ be the error correction vector. We propose to compute $\Delta\hat{X}_{t+1:t+L}$ using a neural network, namely Universal Error Corrector (UEC), which is trained to minimize the error between the corrected values and the ground-truth values. Concretely, the UEC takes the past time-series window and the forecaster's predictions as input and computes the error correction vector. First, using the AR process in Eq. 3, we derive the whole predictions $\hat{X}_{t+1:t+T}$. Next, we iteratively generate the corrections. Formerly, at $k = 0$:

$$\Delta\hat{X}_{t+1:t+L} = \text{UEC}(X_{t-W+1:t}, \hat{X}_{t+1:t+L}) \tag{4}$$

For subsequent AR steps ($k \geq 1$), we compute the correction vectors as:

$$\Delta\hat{X}_{\tau+1:\tau+L} = \text{UEC}(\hat{X}_{\tau-W+1:\tau}, \hat{X}_{\tau+1:\tau+L}) \tag{5}$$

Finally, the whole correction vector $\Delta\hat{X}_{t+1:t+T} = \{\Delta\hat{X}_{t+1}, \Delta\hat{X}_{t+2}, \ldots, \Delta\hat{X}_{t+T}\} \in \mathbb{R}^{T \times D}$ is applied to the forecasted values as follows:

$$\hat{X}_{t+j}^{\text{corr}} = \hat{X}_{t+j} + \beta\Delta\hat{X}_{t+j}, \quad \text{for each} \quad j \in [1, T] \tag{6}$$

where $\beta \in [0, 1]$ is a scalar hyperparameter that controls the strength of the correction. Setting $\beta = 0$ disables the correction entirely, while $\beta = 1$ applies full correction.

**Training Data Preparation**  To train the UEC, we construct supervised training examples where each sample consists of the input $\in \mathbb{R}^{(W+L) \times D}$ to the UEC and its corresponding ground-truth output $\in \mathbb{R}^{L \times D}$. To better reflect realistic deployment scenarios where the forecaster $F$ is likely to produce imperfect predictions, we avoid using the time series used to train $F$, which may lead to overfitted predictions and artificially small errors. Instead, we sample from a held-out validation set, which more accurately represents the model's generalization behavior.

Specifically, we construct training examples for UEC by sampling time series from the validation dataset $\mathcal{D}_{val} = \{X_t\}_{t=T_{train}}^{T_{train}+T_{val}}$. First, we sample a historical window $X_{t-W+1:t}$ of length $W$, along with a corresponding future window $X_{t+1:t+T'} = \{X_{t+1}, X_{t+2}, \ldots, X_{t+T'}\}$, where $T' \geq L$ is a predefined prediction horizon used for training, which can be different than the horizon $T$ during inference. Then, the forecaster $F$ is used to generate the predictions using AR:

$$\hat{X}_{t+1:t+T'} = F_{AR}(X_{t-W+1:t}|T') \tag{7}$$

Next, we sample the ground-truth values $X_{\tau+1:t+(k+1)L} \subseteq X_{t+1:t+T'}$, and compute the ground-truth correction vector as the error between the predicted and the ground-truth time series:

$$\Delta X_{\tau+1:\tau+L} = X_{\tau+1:\tau+L} - \hat{X}_{\tau+1:\tau+L} \tag{8}$$

A training instance for UEC is then a tuple: $\left( \underbrace{(\hat{X}_{\tau-W+1:\tau},\ \hat{X}_{\tau+1:\tau+L})}_{\text{input}},\quad \underbrace{\Delta X_{\tau+1:\tau+L}}_{\text{output}} \right)$

**Standard Training Procedure** We split the $\mathcal{D}_{val}$ data into a training set $\mathcal{U}_{train}$, where the UEC is trained by minimizing a correction loss using the Adam optimizer, and a validation set $\mathcal{U}_{val}$ used for early stopping evaluation. At each iteration, we sample tuples $\left((\hat{X}_{\tau-W+1:\tau},\ \hat{X}_{\tau+1:\tau+L}),\ \Delta X_{\tau+1:\tau+L}\right)$, predict corrections $\Delta\hat{X} = \text{UEC}(\hat{X}_{\tau-W+1:\tau},\ \hat{X}_{\tau+1:\tau+L})$, apply them as:

$$\hat{X}_{\tau+1:\tau+L}^{\text{corr}} = \hat{X}_{\tau+1:\tau+L} + \Delta\hat{X}, \tag{9}$$

and compute the correction loss:

$$\mathcal{L}_{\text{UEC}} = \frac{1}{L} \sum_{j=1}^{L} l_{ec}\left(\hat{X}_{\tau+j}^{\text{corr}},\ X_{\tau+j}\right), \tag{10}$$

where $l_{ec}$ can be any regression loss function, such as MSE or Huber loss. Gradients are backpropagated only through the UEC, keeping the forecaster fixed.

**On Choosing the Correction Strength** To select the correction strength $\beta$ automatically, we propose a balanced validation strategy. We use the validation set $\mathcal{U}_{val}$ that is unseen by both the forecaster $F$ and the UEC, and randomly sample data from the training set $\mathcal{D}_{train}$, denoted $\mathcal{D}_s$, which the forecaster has seen, such that the combined size satisfies $|\mathcal{U}_{val}| + |\mathcal{D}_s| = |\mathcal{D}_{val}|$, where $|\cdot|$ denotes the number of samples in a dataset. This approach prevents bias in either direction: if $\beta$ is tuned only on unseen data, the UEC becomes overly pessimistic about the performance of the forecaster $F$ and selects a high correction strength, which can apply excessive adjustments; if tuned only on seen data, the UEC is too optimistic and selects a low strength. Combining both better reflects realistic deployment conditions, where the forecaster encounters both familiar and unfamiliar data. Additionally, we select separate $\beta$ values depending on the optimization objective: one for MSE and one for MAE, depending on which metric we aim to optimize for in the backbone forecaster $F$.

## 2.3 SEASONAL–TREND UEC ARCHITECTURE

While the UEC can be instantiated with any prediction model, we design an architecture specialized for time-series data by explicitly modeling seasonal and trend components.

**Seasonal–Trend Decomposition.** Given the UEC input $(\hat{X}_{\tau-W+1:\tau},\ \hat{X}_{\tau+1:\tau+L})$, we decompose the backbone prediction part $\hat{X}_{\tau+1:\tau+L}$ into trend and seasonal components:

$$\hat{X}^{\text{t}} = \text{MA}(\hat{X}_{\tau+1:\tau+L}),\ \hat{X}^{\text{s}} = \hat{X}_{\tau+1:\tau+L} - \hat{X}^{\text{t}} \tag{11}$$

where $\text{MA}(\cdot)$ denotes a moving-average filter. We decompose the backbone prediction into seasonal and trend components because time-series data usually exhibit both long-term trends and short-term seasonality. Since the backbone forecaster $F$ may struggle more with one component than the other; explicitly modeling this structure allows UEC to apply targeted corrections.

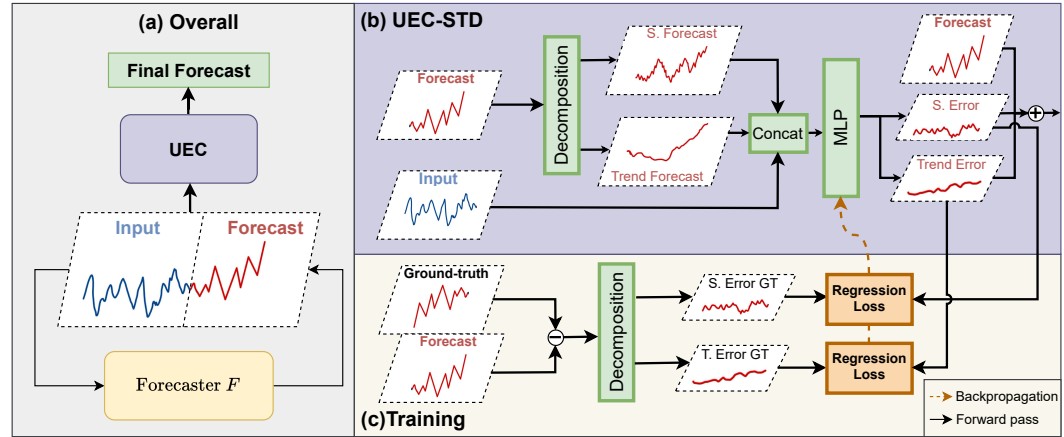

Figure 2: UEC-STD: the corrector refines a pre-trained forecaster by decomposing both the forecast and its error into trend and seasonal components and applying component-wise corrections. *(a) Overall UEC framework:* the corrector takes the input and the forecasted time series from a pre-trained forecaster $F$, and outputs a corrected forecast. *(b) UEC-STD architecture:* the backbone forecast is decomposed into trend and seasonal components, which are concatenated with historical inputs and fed into an MLP to produce separate correction vectors for trend and seasonality. They are summed with the original forecast to make the final forecast. *(c) Training phase:* the ground-truth error is computed as the difference between the forecast and the true values, then decomposed into trend and seasonal error ground-truth components (T. Error GT and S. Error GT) to supervise the corresponding correction outputs.

Next, we fit $\hat{X}^{\mathrm{t}}$ and $\hat{X}^{\mathrm{s}}$ together with the input $\hat{X}_{\tau-W+1:\tau}$ into a multi-layer perceptron (MLP) to produce seasonal and trend correction vectors:

$$\Delta\hat{X}^{\mathrm{t}}, \Delta\hat{X}^{\mathrm{s}} = \mathrm{FF}_\theta\left( \hat{X}_{\tau-W+1:\tau}, \hat{X}^{\mathrm{t}}, \hat{X}^{\mathrm{s}} \right) \tag{12}$$

where $\mathrm{FF}_\theta$ denotes a feed-forward neural network parameterized by $\theta$, and both outputs $\in \mathbb{R}^{L \times D}$.

**Seasonal–Trend Correction.**    The corrected forecast is reconstructed by adjusting each component and summing:

$$\hat{X}^{\mathrm{corr}}_{\tau+1:\tau+L} = \hat{X}_{\tau+1:\tau+L} + \Delta\hat{X}^{\mathrm{t}} + \Delta\hat{X}^{\mathrm{s}} \tag{13}$$

**Seasonal–Trend Training.**    The corresponding ground truth correction vector $\Delta X_{\tau+1:\tau+L}$ is decomposed into:

$$\Delta X^{\mathrm{t}} = \mathrm{MA}(\Delta X_{\tau+1:\tau+L}), \ \Delta X^{\mathrm{s}} = \Delta X_{\tau+1:\tau+L} - \Delta X^{\mathrm{t}} \tag{14}$$

The UEC parameters $\theta$ are learned by minimizing:

$$\mathcal{L}^{st}_{\mathrm{UEC}} = \lambda_{\mathrm{t}}\, l_{ec}\left(\Delta\hat{X}^{\mathrm{t}}, \Delta X^{\mathrm{t}}\right) + \lambda_{\mathrm{s}}\, l_{ec}\left(\Delta\hat{X}^{\mathrm{s}}, \Delta X^{\mathrm{s}}\right), \tag{15}$$

where $\lambda_{\mathrm{t}}$ and $\lambda_{\mathrm{s}}$ control the trade-off between trend and seasonal losses. We refer to this variant as UEC with Seasonal–Trend Decomposition (UEC-STD) to distinguish it from the general UEC.

## 3    EXPERIMENTAL SETUP

**Implementation**    We conducted experiments using a standard time-series benchmark and codebase[1]. Initially, we trained the backbone forecaster using the normal codebase training, with the MSE as the loss function $l_{fc}$. The specific hyperparameters used for training are consistent with established best practices in the field. For example, we fix the batch size to 128, the learning rate

---

[1] https://github.com/thuml/Time-Series-Library

to 0.01, and use the Adam optimizer with default parameters ($\beta_1 = 0.9, \beta_2 = 0.999, \epsilon = 10^{-8}$), and train for 10 epochs with early stopping patience of 10. For further details on the exact parameter settings, we refer the reader to the official codebase. This trained backbone was then used to generate data for the training of the UEC. For UEC, we found that using $l_{ec}$ as the Huber loss led to more stable training for the UEC (see Sec. 4.3), and we therefore adopted it for all subsequent experiments. More details on UEC hyperparameters can be found in Appendix A.

**Computing Requirement**   All experiments are conducted on a single NVIDIA V100 GPU. The training cost of the proposed UEC modules is negligible compared to that of the backbone models. For example, training the TimeMixer backbone on ETTh1 with $L \in [96, 192, 336, 720]$ requires approximately 10 minutes of GPU time, whereas training UEC-STD on that setting takes only about 1 minute, i.e., roughly one-tenth of the backbone training time. This demonstrates that our approach introduces minimal computational overhead while maintaining efficiency.

**Evaluation Protocol**   For each dataset and prediction length $L$, we (i) train the backbone forecaster on the standard training split (70%) and use the validation split to get the best checkpoint, (ii) train the UEC on the validation split (10%) to correct the backbone, and (iii) report results on the held-out test split (20%). We report average Mean Squared Error (MSE) and Mean Absolute Error (MAE):

$$\text{MSE} = \frac{1}{NLD} \sum_{i=1}^{N} \sum_{j=1}^{L} \sum_{d=1}^{D} \left( \hat{X}_{t+j,d}^{(i)} - X_{t+j,d}^{(i)} \right)^2, \text{ MAE} = \frac{1}{NLD} \sum_{i=1}^{N} \sum_{j=1}^{L} \sum_{d=1}^{D} \left| \hat{X}_{t+j,d}^{(i)} - X_{t+j,d}^{(i)} \right|$$

Here $N$ is the number of test segments, $L$ the forecast horizon, and $D$ the dimensionality. We compute metrics per prediction length and then take the mean across lengths.

## 4   EXPERIMENTAL RESULTS

This section aims to demonstrate the effectiveness of our proposed approach for enhancing autoregressive inference in long-term forecasting. We begin by establishing that autoregressive inference is a strong baseline, warranting further investigation for targeted improvements. We then demonstrate that the limitation of AR can be addressed by integrating UEC into the inference pipeline, resulting in significant performance gains across various backbone forecasters. More specifically, we evaluate multiple design choices for UEC and demonstrate that our proposed UEC-STD architecture consistently achieves the best results across all benchmarks. Finally, we conduct ablation studies and model analyses to assess the contribution of each component in our approach.

### 4.1   RESULTS ON TIME-SERIES BENCHMARK

#### AUTOREGRESSION IS A STRONG BASELINE, BUT CORRECTING ITS ERRORS IS NECESSARY

We compare two paradigms for long-term forecasting: (i) **Direct Forecasting (DF)**, which predicts the entire horizon in one pass, and (ii) **Autoregressive (AR)**, which generates predictions iteratively. DF requires horizon-specific models and a higher cost, while AR reuses the same module across steps, making it more efficient and flexible. Experiments on ETTh1, Weather, and Electricity with two backbones (TimeMixer (Wang et al., 2024a) and TimesNet (Wu et al., 2023)) show that AR matches or outperforms DF in 7 of 12 cases (Appendix Table 13), particularly excelling on ETTh1. However, AR suffers from *error accumulation*, where small early mistakes amplify into high MSE/MAE (0.4–0.7) over long horizons, corresponding to up to 28.8% error increase compared to using ground-truth inputs (Fig. 1 (b)). This underscores the need for error correction. Hence, we focus on AR as the main target for correction and omit the DF baseline to save computation.

#### UEC-STD DELIVERS SUBSTANTIAL AND CONSISTENT IMPROVEMENTS TO AR

The purpose of this experiment section is to evaluate the effectiveness of our proposed UEC in mitigating the errors and improving the overall performance of modern deep forecasting models under autoregressive inference. As such, we examine different UEC architectures on **3 forecasting backbones** (*TimeMixer* (Wang et al., 2024a), *TimesNet* (Wu et al., 2023), and *TimeXer* (Wang et al.,

Table 1: Average Error Reduction in MSE compared to backbone for different UEC methods (the lower the better, negative means improvement). N/A indicates that the method failed to converge or crashed during training. Bold and underline denote best and second-best results, respectively.

| Method | ETTh1 | ETTh2 | ETTm1 | ETTm2 | Traffic | Weather | Electricity |
|---|---|---|---|---|---|---|---|
| AR (No Correction) | 0.00 | 0.00 | 0.00 | 0.00 | 0.00 | 0.00 | 0.00 |
| UEC-MLP | 0.71 | 0.05 | -0.93 | -1.20 | -0.67 | -1.34 | 0.17 |
| UEC-Logistic | 11.7 | 5.84 | -3.49 | 0.25 | N/A | **-3.61** | N/A |
| UEC-Random Forest | 1.10 | -1.39 | -0.92 | -1.43 | N/A | 0.76 | N/A |
| UEC-XGBoost | 0.40 | -0.46 | **-11.88** | -0.51 | N/A | -2.48 | N/A |
| UEC-LSTM | 2.48 | -0.08 | -0.29 | 24.63 | 0.36 | 6.35 | -0.52 |
| UEC-GRU | 3.49 | -0.51 | -0.29 | 4.32 | -1.12 | 4.13 | -0.26 |
| UEC-CNN | 0.94 | -0.77 | -0.76 | 1.96 | 0.06 | 4.99 | 0.04 |
| UEC-Transformer | 0.91 | -1.22 | -0.63 | 0.47 | -0.18 | -1.66 | **-1.19** |
| UEC-STD | **-2.39** | **-1.49** | -4.78 | **-1.78** | **-1.18** | -2.10 | -0.91 |

2024b)). They are chosen as efficient and recent strong baselines in time-series long-term forecasting. We select **7 datasets** (ETTh1, ETTh2, ETTm1, ETTm2, Electricity, Weather, and Traffic), which support a long-term prediction horizon up to 720 steps. Moreover, we evaluate **9 different UEC architectures**, ranging from classic machine learning models such as logistic regression and random forests, to simple neural networks like MLPs and LSTMs, and more sophisticated models such as Transformers. These architectures follow the standard UEC framework (Sec. 2.2). We denote these methods as UEC-X, where X refers to the underlying correcting architecture (see Appendix B). We also include the proposed UEC-STD variant (Sec. 2.3) to validate our special design for time-series data. All UEC methods apply auto selection of $\beta$ (Sec. 2.2). To see how UEC helps the forecasters, we report the error reduction rate (%, Appendix Eq. 18) in MSE and MAE for various UEC architectures compared to no correction ($\beta = 0$). The error reduction is then averaged over 3 backbones. Negative values indicate an improvement over the backbone model with no correction, while positive values denote performance degradation.

Table 1 and Appendix Table 3 summarize the results for improvements in MSE and MAE, respectively. Regarding MSE, overall, most architectures, particularly XGBoost and UEC-STD, achieve consistent error reductions across multiple datasets. However, some classical machine learning models, such as XGBoost, Random Forest, and Logistic Regression, fail to scale effectively on large, high-dimensional datasets like Traffic and Electricity, resulting in training convergence issues despite extensive hyperparameter tuning. Therefore, UEC-STD achieves the best overall performance, delivering both the greatest average error reduction and the highest consistency across datasets. In terms of MAE, UEC-STD is the only method that can reliably correct the forecaster's errors. On average, across backbones and datasets, **UEC-STD achieves MSE and MAE improvements of 2.1% and 0.8%, respectively**, which is comparable to SOTA improvements in the field (Wang et al., 2024b). Notably, for datasets like ETTm1, UEC-STD attains major error reductions of 4.78% in MSE and 1.81% in MAE. We provide the details of these experimental results in Appendix C.

## 4.2 ABLATION STUDY ON UEC-STD

**Seasonal-Trend Decomposition Components** Here, we compare different design choices for seasonal–trend decomposition (STD) by varying the choice of STD components in UEC input and output (Table 2). We observe that adding trend or seasonal components to inputs only (*No STD Output*) yields little improvement compared to not using STD at all (*No STD*), with gains of 1.1% MSE on ETTh1 and 0.4% MSE on Weather, while Traffic shows no change. Modeling STD in UEC output further improves the performance. In particular, when predicting only seasonal (*No Trend Output*) or only trend (*No Seasonal Output*), we find that seasonal correction contributes more to ETTh1 (seasonal-only improves MSE by 5.3% vs trend-only 2.7%), whereas both Traffic and Weather exhibit little to no improvement when relying on only one component. Our full setup (*Full*), which uses both decomposed inputs and predicts separate errors for trend and seasonal components, achieves the best overall performance, improving MSE/MAE by 5.99%/2.48% on ETTh1, 0.37%/0.30% on Traffic, and 0.83%/1.45% on Weather compared to the No STD. These demonstrate the complementary benefits of jointly correcting trend and seasonality, leading to consistent gains across datasets.

Table 2: Comparison of different design variants for seasonal–trend decomposition (STD). Each setting differs in the choice of inputs (raw series $\hat{X}$, seasonal $\hat{X}^s$, trend $\hat{X}^t$) and outputs (predicted errors $\Delta\hat{X}$, $\Delta\hat{X}^s$, $\Delta\hat{X}^t$). Bold denotes the best results.

| Setting | Input(s) | Output(s) | ETTh1 | | Traffic | | Weather | |
| --- | --- | --- | --- | --- | --- | --- | --- | --- |
| | | | MSE | MAE | MSE | MAE | MSE | MAE |
| No STD | $\hat{X}_{\tau-W+1:\tau}$ | $\Delta\hat{X}$ | 0.451 | 0.444 | 0.545 | 0.336 | 0.241 | 0.276 |
| No STD Output | $\hat{X}_{\tau-W+1:\tau}, \hat{X}^t, \hat{X}^s$ | $\Delta\hat{X}$ | 0.446 | 0.452 | 0.546 | 0.336 | 0.240 | 0.272 |
| No Season Output | $\hat{X}_{\tau-W+1:\tau}, \hat{X}^t, \hat{X}^s$ | $\Delta\hat{X}^t$ | 0.464 | 0.447 | 0.544 | 0.336 | 0.245 | 0.283 |
| No Trend Output | $\hat{X}_{\tau-W+1:\tau}, \hat{X}^t, \hat{X}^s$ | $\Delta\hat{X}^s$ | 0.427 | 0.437 | 0.547 | 0.338 | 0.244 | 0.276 |
| Full (Our) | $\hat{X}_{\tau-W+1:\tau}, \hat{X}^t, \hat{X}^s$ | $\Delta\hat{X}^t, \Delta\hat{X}^s$ | **0.424** | **0.433** | **0.543** | **0.335** | **0.239** | **0.272** |

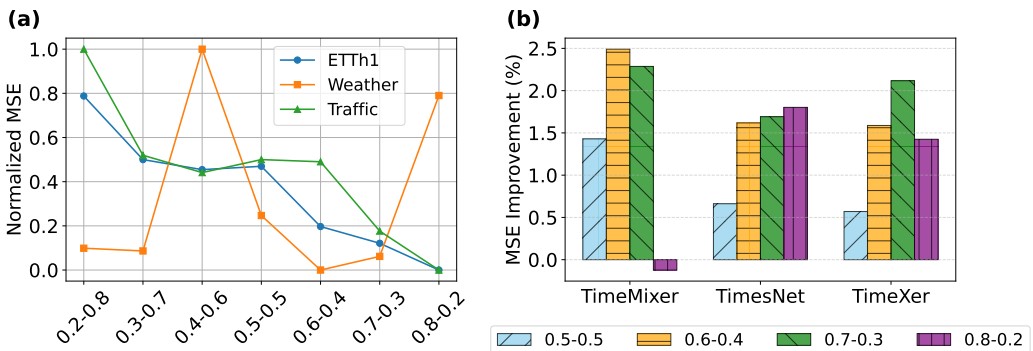

Figure 3: Seasonal-Trend (ST) Coefficient $\lambda_s$–$\lambda_t$ analysis. (a) Normalized MSE (0–1) for 3 datasets, ETTh1, Weather, and Traffic, using TimeMixer across different coefficients; lower values indicate better performance. (b) Percentage MSE improvement on Weather for three backbones (TimeMixer, TimesNet, TimeXer) with varying ST coefficients; higher values indicate greater improvement. The plots show how emphasizing the correction of seasonal and trend affects forecasting performance.

**Seasonal–Trend Coefficients** We study different seasonal–trend (ST) coefficient settings $\lambda_s$–$\lambda_t$ across datasets (ETTh1, Weather, Traffic) and backbones (TimeMixer, TimesNet, TimeXer). In Fig.3a, we fix $\beta = 0.1$ and vary coefficients from 0.2–0.8 to 0.8–0.2. Results show that higher seasonal weighting improves accuracy: ETTh1 and Traffic perform best with 0.8–0.2 (1.4%, 0.96% improvements), while Weather prefers 0.6–0.4 (1.07%). Overweighting seasonality, however, can hurt datasets dominated by long-term trends. In Fig. 3b, we repeat this analysis for Weather across backbones. The trend holds broadly: emphasizing seasonality improves accuracy, though the optimal balance depends on datasets and backbones. Overall, we recommend starting with 0.5–0.5 and adjusting toward seasonality (e.g., 0.6–0.4 or 0.8–0.2) based on dataset characteristics.

## 4.3 MODEL ANALYSIS

In this section, we analyze the general behavior of the UEC framework. For simplicity and to reduce the confounding effects of Seasonal–Trend components, we use UEC-MLP as the representative architecture, while we expect UEC-STD to exhibit better behaviors.

**Long-term Correction Behaviors** We present four qualitative cases in Appendix Fig. 4 comparing predictions *with* and *without* UEC on the *Traffic* dataset (prediction length = 720). Across all cases, the UEC-enhanced forecasts closely follow the ground truth in level, trend, and oscillation, whereas the no-UEC baseline exhibits *collapse*, which shows nearly flat, low-variance trajectories that remain anchored to early forecast values. In general, UEC helps long-horizon rollouts by adding learned, context-aware residuals to the backbone forecast at each autoregressive step. These corrections restore amplitude and phase, counter drift, and smooth chunk boundaries, so predictions maintain appropriate variability and stay aligned with the target signal.

**UEC Training Loss**    To examine the impact of training loss on UEC performance, we report results using different $l_{ec}$ (Huber, L1, and MSE) in Appendix Table 14. Experiments use ETTh1 dataset with 2 backbones: TimeMixer and TimesNet. Overall, Huber loss achieves the lowest average MSE and MAE in four cases, the best among the three losses. While different losses may yield gains in other cases, we adopt Huber loss as the default for training UEC to avoid costly tuning.

**Improvement Gain with Extended Training.**    One question is whether UEC's gains arise from holding out validation data for training the corrector. To test this, we retrain backbones on both training and validation sets (so UEC has no data advantage) and then train UEC on the same validation portion to correct the new backbones. Results on Traffic (Appendix Fig. 6) show UEC still improves performance, confirming the benefits come from learning correction patterns rather than data withholding. Improvements vary by backbone: weaker models like TimesNet gain more, while stronger ones like TimeMixer benefit less and may even overfit when retrained with extra data. Hence, we recommend training backbones on the original data and reserving validation solely for UEC.

## 5   RELATED WORKS

**Classical Error Correction Models**    Traditional Error Correction Models (ECMs) are widely used in econometrics (Hansen, 2003; Barigozzi et al., 2024). These models explicitly capture deviations from equilibrium and apply corrective terms to guide predictions back toward the expected state. However, ECMs are designed for linear, low-dimensional systems and rely on statistical assumptions that are difficult to transfer to the complex dynamics of modern deep-learning models. Their reliance on multivariate co-integration prevents their applicability to high-dimensional forecasting scenarios.

**Autoregressive Deep Learning and Error Accumulation**    Deep learning models have recently achieved state-of-the-art performance in time-series forecasting (Liu et al., 2023; Zeng et al., 2023; Wang et al., 2024b). TimeMixer (Wang et al., 2024a), a decomposable multiscale mixing framework, improves forecasting by separating temporal components and mixing information across multiple scales with high efficiency. TimeMixer++ (Wang et al., 2025) further generalizes this approach by introducing a universal time-series pattern machine that enhances multi-scale modeling across diverse predictive tasks. DeformableTST (Luo & Wang, 2024) addresses the limitations of traditional transformer patching by incorporating deformable attention, enabling the model to flexibly focus on the most relevant temporal regions without fixed segmentation. Cross-series relational models like TimeBridge (Liu et al., 2025) learn dependencies among correlated time series through inter-series attention, leveraging shared patterns to improve multivariate forecasting performance. Despite the advances, these models often train with fixed input-output lengths, and to predict longer horizons, they must rely on autoregressive decoding: using the prediction as the input for the next forecasting step. Unfortunately, this recursive strategy leads to unavoidable compounding errors over longer horizons (Moreno-Pino et al., 2023). A temporary workaround is to train separate models for different prediction lengths. While this can help manage error accumulation, it incurs additional training time, storage, and complexity costs. Thus, it is not suited for ultra-long or unknown inference lengths, limiting its scalability and practical applicability.

**Error Correction in Deep Learning for Time-Series Forecasting**    . Recent studies have explored incorporating error correction mechanisms using deep learning to improve time-series forecasting accuracy. Liu et al. (2020) propose modules that explicitly learn residual errors during training, while Zhang et al. (2021) refine predictions using predefined loss-based error functions. Others attempt to learn the error correction function, such as using LSTMs to model the residuals of classical ARIMA forecasts (Nandutu et al., 2022) or (Li et al., 2024), jointly training the forecasting model with a diffusion process to refine its predictions. While promising, these methods are often tied to specific architectures or training pipelines, limiting their generality. To date, no architecture-agnostic error correction approach consistently improves modern forecasters. This work is the first to address this gap by proposing a general and modular solution.

## 6 CONCLUSION

In this paper, we revisited the problem of error accumulation in deep autoregressive time-series forecasting and proposed a simple, architecture-agnostic error correction mechanism that can be integrated with any existing deep learning forecaster without retraining. Our proposed approach, named Universal Error Corrector with Seasonal-Trend Decomposition (UEC-STD), consistently improves long-term prediction accuracy across multiple benchmarks and backbone models, providing both practical utility and novel insights into autoregressive error mitigation. While effective, our method introduces a modest computational overhead due to the additional error correction prediction. Future work will focus on designing more efficient UEC variants that minimize computational overhead without compromising performance. Moreover, investigating adaptive correction mechanisms and extending our evaluation to diverse real-world scenarios, such as multi-modality and irregularly sampled time series, offers promising avenues to improve the robustness and scalability of deep time-series forecasting.

## REPRODUCIBILITY STATEMENT

Details of implementations and experiments can be found in the Appendix. Upon publication, we will release the implementation as open-source with the necessary instructions to ensure reproducibility.

## LLM USAGE

Large Language Models (LLMs) were not involved in the design, implementation, or analysis of our method. They were only used to refine the presentation of the paper by correcting grammar and improving writing clarity.

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

APPENDIX

# A DETAILS ON UEC-STD IMPLEMENTATIONS

## A.1 TRAINING AND EVALUATION SUMMARY

For each dataset and prediction length $L$, the training and evaluation process consists of four stages:

1. **Backbone training.** The forecaster $F$ is trained on the training split $\mathcal{D}_{train}$ (70%), and the best checkpoint is selected based on performance on the validation split $\mathcal{D}_{val}$ (10%).

2. **UEC-STD training.** Supervised seasonal and trend correction data ($\mathcal{U}$train and $\mathcal{U}$val) is derived from the validation split $\mathcal{D}_{val}$, where 70% is used for training and 30% is reserved for early stopping and tuning the correction strength $\beta$. The UEC-STD is then trained following the procedure described in Sect. 2.2 and Sect. 2.3, using 100 training steps with a batch size of 64.

3. **Correction strength selection.** The correction weight $\beta \in [0, 1]$ is tuned automatically using the validation strategy described in Sect. 2.2.

4. **Evaluation.** The trained UEC-STD is applied autoregressively to backbone forecasts, and corrected predictions are generated according to Eq. 6. Final performance is reported on the held-out test split (20%).

## A.2 SEASONAL–TREND MOVING AVERAGE DECOMPOSITION.

We decompose the backbone forecast $\hat{X}_{\tau+1:\tau+L}$ into trend and seasonal components using moving average decomposition:

$$\hat{X}^{\mathrm{t}} = \mathrm{MA}(\hat{X}_{\tau+1:\tau+L}), \quad \hat{X}^{\mathrm{s}} = \hat{X}_{\tau+1:\tau+L} - \hat{X}^{\mathrm{t}}, \tag{16}$$

where $\mathrm{MA}(\cdot)$ is a 1D convolution-based centred moving average (default kernel size $ks = 25$), computed as in Algorithm 1.

---
**Algorithm 1** 1D Moving-Average Trend Computation

---
1: **Input:** $\hat{X}_{\tau+1:\tau+L}$, kernel size $ks$ (odd, default 25)
2: **Output:** Trend component of $\hat{X}_{\tau+1:\tau+L}$, same shape
3: $pad \leftarrow (ks - 1)/2$
4: $filt \leftarrow$ 1D averaging filter of length $ks$ with values $1/ks$
5: $\hat{X}^{\mathrm{t}} \leftarrow \mathrm{conv1d}(\hat{X}_{\tau+1:\tau+L}, filt, padding = pad)$
6: **Return** $\hat{X}^{\mathrm{t}}$

---

Next, we fit $\hat{X}^{\mathrm{t}}$ and $\hat{X}^{\mathrm{s}}$ together with the input $\hat{X}_{\tau-W+1:\tau}$ into a multi-layer perceptron (MLP) to produce seasonal and trend correction vectors:

$$\widehat{\Delta} X^{\mathrm{t}}, \; \widehat{\Delta} X^{\mathrm{s}} = \mathrm{FF}_\theta\left( \hat{X}_{\tau-W+1:\tau}, \; \hat{X}^{\mathrm{t}}, \; \hat{X}^{\mathrm{s}} \right) \tag{17}$$

## A.3 MODEL ARCHITECTURE

$\mathrm{FF}_\theta$ is a lightweight two-stage MLP designed to refine base predictions by modeling seasonal and trend errors. Assuming an input tensor $x \in \mathbb{R}^{B \times T \times D}$, it will be processed as follows.

Before entering Subnetwork 1, the input $x$ is reshaped to $(B \times D, T)$ so that each feature dimension can be processed independently along the temporal axis. Subnetwork 1 applies a two-layer MLP with ReLU activation and dropout to capture temporal dependencies in a parameter-efficient manner:

**Subnetwork 1:**

$$h = \mathrm{Dropout}\big(W_2\, \sigma(W_1 x)\big),$$

where $W_1 \in \mathbb{R}^{T \times H}, W_2 \in \mathbb{R}^{H \times T}$, $\sigma$ denotes the ReLU activation, and $H$ is the hidden size (default $H = 32$). This design allows the model to capture temporal dependencies in a parameter-efficient manner while using dropout value of $0.5$ for regularization.

The output of Subnetwork 1 is then permuted back to $(B, T, D)$ before entering Subnetwork 2. This second subnetwork is a two-layer MLP, which is responsible for aggregating feature information and projecting into the output space:

**Subnetwork 2:**

$$y = \text{Dropout}\big(W_4 \, \sigma(W_3 h)\big),$$

where $W_3 \in \mathbb{R}^{D \times H}$ and $W_4 \in \mathbb{R}^{H \times D}$. We then split $y$ into $y_{trend} = \widehat{\Delta} X^{\text{t}}$ and $y_{seasonal} = \widehat{\Delta} X^{\text{s}}$ where both $y_{\text{trend}}, y_{\text{seasonal}} \in \mathbb{R}^{B \times L \times D}$. These components are subsequently used in Eq. 13 to compute the final correction value.

# B    DETAILS ON BASELINE IMPLEMENTATIONS

We implement a diverse set of baseline error correctors spanning traditional machine learning approaches and modern neural architectures. Throughout, each of these UEC models takes the input sequence $x = (\hat{X}_{\tau - W + 1 : \tau}, \ \hat{X}_{\tau + 1 : \tau + L})$ where $x \in \mathbb{R}^{B \times T \times D}$ and outputs $y = \Delta \hat{X}_{\tau + 1 : \tau + L}$ where $y \in \mathbb{R}^{B \times L \times D}$. These baseline correctors were also trained on the correction data constructed from the validation split $\mathcal{D}_{val}$, similar to our proposed UEC-STD.

## B.1    TRADITIONAL MODELS

**UEC-Logistic.**    We implement a logistic regression model using `scikit-learn`'s pipeline (Pedregosa et al., 2011), which combines feature scaling, PCA, and a ridge regression head. Specifically, $x$ is flattened into $(B, T \times D)$, normalized via `StandardScaler`, reduced using PCA to retain 95% of variance, and finally fitted with a ridge regressor using the SAG solver to predict flattened targets $(B, L \times D)$. The predicted output is then reshaped back to $(B, L, D)$ to match the original temporal and feature dimensions.

**UEC-Random Forest.**    A random forest regressor using `scikit-learn` (Pedregosa et al., 2011) is trained on flattened features $(B, T \times D)$ to predict flattened targets $(B, L \times D)$. We use 20 trees with a maximum depth of 6. The predicted outputs are reshaped back to $(B, L, D)$ to recover the original temporal structure.

**UEC-XGBoost.**    We implement an XGBoost regressor with GPU acceleration (`tree_method=gpu_hist`, `device=cuda`) using `dmlc xgboost.XGB` (Chen & Guestrin, 2016). Similar to Random Forest, $x$ is flattened into $(B, T \times D)$. The default configuration uses 20 boosting rounds, maximum depth 6, learning rate 0.3, and subsample ratio 1.0. After prediction, outputs are reshaped from $(B, L \times D)$ back to $(B, L, D)$ to maintain consistency with the input dimensions.

## B.2    NEURAL MODELS

**UEC-MLP.**    As a simple neural baseline, we uses the same architecture as described in Sect. A.3, but directly takes the original forecast $\hat{X}_{\tau + 1 : \tau + L}$ as input without decomposing it into trend and seasonal components.

**UEC-LSTM & UEC-GRU.**    We implement both GRU- and LSTM-based recurrent correctors. Given $x \in \mathbb{R}^{B \times T \times D}$, the sequence is passed through an RNN encoder (hidden dimension 32, configurable layers, dropout 0.5). The hidden outputs $(B, T, H)$ are projected through a two-layer MLP with ReLU activations and dropout to produce $(B, L, D)$.

**UEC-CNN.**    We apply 1D temporal convolutions to capture local dependencies in the sequence. The input $x$ is permuted to $(B, D, T)$ and processed by two convolutional layers (kernel size 3, hidden dimension 32), followed by dropout. The output is projected with a two-layer MLP into $(B, L, D)$.

**UEC-Transformer.** We use a transformer encoder with learnable positional embeddings. The input $x$ is first projected into a hidden space ($64$ dimensions), added with positional encodings, and passed through a stack of $2$ encoder layers with $4$ attention heads and feedforward dimension $128$. The outputs are mapped via a two-layer MLP with ReLU and dropout to $(B, L, D)$.

## B.3 TRAINING SETUP

Each baseline is evaluated under the same autoregressive correction setting as our proposed model for fair comparison.

# C DETAILS ON EXPERIMENTAL RESULTS

## C.1 EVALUATION METRIC

The reduction is calculated as:

$$\text{Error Reduction} = \frac{\text{MSE/MAE}_{\text{UEC}} - \text{MSE/MAE}_{\text{Backbone}}}{\text{MSE/MAE}_{\text{Backbone}}} \times 100\% \tag{18}$$

## C.2 AVERAGE MAE REDUCTION ACROSS MODELS

Table 3 reports the average error reduction in MAE compared to the backbone for different UEC methods. Negative values indicate improvements, while positive values denote error increases. N/A indicates that the method failed to converge or crashed during training. Bold and underline denote best and second-best results, respectively.

Table 3: Average Error Reduction in MAE compared to backbone for different UEC methods (the lower the better, negative means improvement). N/A indicates that the method failed to converge or crashed during training. Bold and underline denote best and second-best results, respectively.

| Method | ETTh1 | ETTh2 | ETTm1 | ETTm2 | Traffic | Weather | Electricity |
|---|---|---|---|---|---|---|---|
| AR (No Correction) | 0.00 | 0.00 | 0.00 | 0.00 | 0.00 | 0.00 | 0.00 |
| UEC-MLP | 0.01 | 0.21 | -0.48 | -0.31 | -1.09 | 2.20 | -0.08 |
| UEC-Logistic | 0.91 | 9.06 | -0.97 | 1.02 | N/A | 2.54 | N/A |
| UEC-Random Forest | **-0.74** | -0.48 | -1.27 | **-1.05** | N/A | 3.51 | N/A |
| UEC-XGBoost | -0.47 | 0.85 | **-5.72** | 0.42 | N/A | 4.01 | N/A |
| UEC-LSTM | 2.25 | 0.13 | -0.20 | 14.5 | **-1.70** | 3.72 | -0.48 |
| UEC-GRU | 3.53 | 0.30 | -0.26 | 3.05 | -1.53 | 3.04 | -0.32 |
| UEC-CNN | 1.99 | -0.33 | 0.17 | 1.19 | -0.43 | 1.24 | -0.13 |
| UEC-Transformer | 0.90 | -0.24 | -0.39 | 7.45 | -0.82 | 1.39 | **-1.09** |
| UEC-STD | -0.44 | **-0.50** | -1.81 | -0.50 | -0.89 | **-0.83** | -0.85 |

## C.3 RAW MSE AND MAE RESULTS

Table 4, Table 5 and Table 7 report the raw MSE and MAE results for all compared methods under the TimeMixer, TimesNet and TimeXer backbones, respectively. For each dataset and prediction horizon, the best and second-best values are highlighted in red and blue. The bottom rows further summarize the number of times each method achieved the best or second-best performance across all settings. These results form the basis for the error-reduction analyses in the main text and clearly demonstrate that our proposed UEC-STD consistently delivers the best overall performance.

Table 4: Raw MSE and MAE results using TimeMixer as the backbone forecaster across multiple datasets and horizons. Lower values are better. Red denotes the best value and blue is the second best.

| Dataset | | STD (Ours) | | MLP | | Logistic | | RF | | XGB | | LSTM | | GRU | | CNN | | TF. | | TimeMixer | |
|---|---|---|---|---|---|---|---|---|---|---|---|---|---|---|---|---|---|---|---|---|---|
| | | MSE | MAE | MSE | MAE | MSE | MAE | MSE | MAE | MSE | MAE | MSE | MAE | MSE | MAE | MSE | MAE | MSE | MAE | MSE | MAE |
| ETTh1 | 96 | 0.370 | 0.399 | 0.393 | 0.407 | 0.381 | 0.402 | 0.392 | 0.3400 | 0.388 | 0.397 | 0.378 | 0.404 | 0.383 | 0.426 | 0.376 | 0.408 | 0.387 | 0.408 | 0.377 | 0.397 |
| | 192 | 0.414 | 0.425 | 0.440 | 0.436 | 0.427 | 0.430 | 0.437 | 0.428 | 0.433 | 0.426 | 0.428 | 0.432 | 0.433 | 0.454 | 0.424 | 0.436 | 0.437 | 0.437 | 0.427 | 0.427 |
| | 336 | 0.449 | 0.444 | 0.475 | 0.456 | 0.464 | 0.451 | 0.470 | 0.448 | 0.469 | 0.447 | 0.470 | 0.455 | 0.462 | 0.458 | 0.479 | 0.459 | 0.465 | 0.449 | 0.465 | 0.449 |
| | 720 | 0.463 | 0.463 | 0.496 | 0.480 | 0.480 | 0.470 | 0.476 | 0.464 | 0.484 | 0.464 | 0.482 | 0.475 | 0.491 | 0.499 | 0.475 | 0.481 | 0.500 | 0.480 | 0.474 | 0.466 |
| | Avg | 0.424 | 0.433 | 0.451 | 0.445 | 0.438 | 0.438 | 0.444 | 0.420 | 0.444 | 0.434 | 0.440 | 0.442 | 0.445 | 0.464 | 0.434 | 0.446 | 0.451 | 0.459 | 0.435 | 0.434 |
| ETTh2 | 96 | 0.292 | 0.343 | 0.293 | 0.344 | 0.326 | 0.399 | 0.290 | 0.343 | 0.294 | 0.350 | 0.296 | 0.346 | 0.293 | 0.347 | 0.294 | 0.344 | 0.291 | 0.344 | 0.293 | 0.343 |
| | 192 | 0.374 | 0.395 | 0.377 | 0.396 | 0.410 | 0.447 | 0.371 | 0.394 | 0.375 | 0.400 | 0.377 | 0.396 | 0.377 | 0.395 | 0.377 | 0.395 | 0.371 | 0.394 | 0.376 | 0.395 |
| | 336 | 0.427 | 0.437 | 0.431 | 0.440 | 0.463 | 0.487 | 0.422 | 0.435 | 0.428 | 0.443 | 0.428 | 0.439 | 0.424 | 0.439 | 0.430 | 0.438 | 0.422 | 0.436 | 0.428 | 0.438 |
| | 720 | 0.510 | 0.492 | 0.513 | 0.496 | 0.556 | 0.540 | 0.497 | 0.485 | 0.508 | 0.495 | 0.512 | 0.496 | 0.507 | 0.494 | 0.504 | 0.490 | 0.499 | 0.488 | 0.510 | 0.493 |
| | Avg | 0.401 | 0.416 | 0.404 | 0.419 | 0.439 | 0.468 | 0.395 | 0.414 | 0.401 | 0.422 | 0.403 | 0.419 | 0.399 | 0.419 | 0.401 | 0.417 | 0.396 | 0.416 | 0.402 | 0.417 |
| ETTm1 | 96 | 0.318 | 0.362 | 0.325 | 0.360 | 0.322 | 0.360 | 0.326 | 0.361 | 0.321 | 0.361 | 0.327 | 0.362 | 0.328 | 0.362 | 0.326 | 0.367 | 0.291 | 0.344 | 0.293 | 0.343 |
| | 192 | 0.374 | 0.396 | 0.385 | 0.397 | 0.379 | 0.396 | 0.385 | 0.399 | 0.378 | 0.397 | 0.387 | 0.399 | 0.386 | 0.400 | 0.386 | 0.403 | 0.388 | 0.400 | 0.388 | 0.400 |
| | 336 | 0.425 | 0.428 | 0.440 | 0.432 | 0.433 | 0.431 | 0.440 | 0.434 | 0.431 | 0.431 | 0.442 | 0.434 | 0.443 | 0.435 | 0.440 | 0.437 | 0.443 | 0.435 | 0.443 | 0.436 |
| | 720 | 0.546 | 0.484 | 0.568 | 0.492 | 0.558 | 0.591 | 0.569 | 0.495 | 0.554 | 0.490 | 0.573 | 0.495 | 0.570 | 0.496 | 0.573 | 0.496 | 0.573 | 0.496 | 0.575 | 0.498 |
| | Avg | 0.416 | 0.418 | 0.430 | 0.420 | 0.423 | 0.445 | 0.430 | 0.422 | 0.421 | 0.420 | 0.432 | 0.423 | 0.434 | 0.423 | 0.431 | 0.426 | 0.424 | 0.419 | 0.423 | 0.419 |
| ETTm2 | 96 | 0.174 | 0.259 | 0.174 | 0.258 | 0.173 | 0.267 | 0.171 | 0.259 | 0.173 | 0.266 | 0.185 | 0.276 | 0.185 | 0.276 | 0.202 | 0.289 | 0.175 | 0.258 | 0.176 | 0.258 |
| | 192 | 0.242 | 0.303 | 0.243 | 0.303 | 0.238 | 0.308 | 0.235 | 0.302 | 0.237 | 0.308 | 0.253 | 0.321 | 0.253 | 0.321 | 0.267 | 0.330 | 0.242 | 0.303 | 0.245 | 0.304 |
| | 336 | 0.310 | 0.345 | 0.312 | 0.347 | 0.303 | 0.350 | 0.299 | 0.344 | 0.300 | 0.349 | 0.321 | 0.364 | 0.321 | 0.364 | 0.331 | 0.370 | 0.310 | 0.347 | 0.316 | 0.349 |
| | 720 | 0.419 | 0.408 | 0.422 | 0.411 | 0.407 | 0.410 | 0.405 | 0.406 | 0.405 | 0.410 | 0.427 | 0.424 | 0.427 | 0.424 | 0.431 | 0.427 | 0.418 | 0.410 | 0.427 | 0.413 |
| | Avg | 0.288 | 0.328 | 0.288 | 0.329 | 0.280 | 0.334 | 0.278 | 0.327 | 0.279 | 0.334 | 0.322 | 0.346 | 0.322 | 0.346 | 0.308 | 0.342 | 0.286 | 0.329 | 0.290 | 0.329 |
| Traffic | 96 | 0.477 | 0.310 | 0.478 | 0.310 | N/A | N/A | N/A | N/A | N/A | N/A | 0.476 | 0.308 | 0.477 | 0.309 | 0.480 | 0.311 | 0.481 | 0.311 | 0.481 | 0.312 |
| | 192 | 0.514 | 0.323 | 0.515 | 0.322 | N/A | N/A | N/A | N/A | N/A | N/A | 0.513 | 0.320 | 0.513 | 0.321 | 0.518 | 0.324 | 0.519 | 0.324 | 0.518 | 0.325 |
| | 336 | 0.554 | 0.337 | 0.556 | 0.337 | N/A | N/A | N/A | N/A | N/A | N/A | 0.552 | 0.335 | 0.553 | 0.336 | 0.560 | 0.340 | 0.560 | 0.340 | 0.560 | 0.340 |
| | 720 | 0.627 | 0.372 | 0.631 | 0.374 | N/A | N/A | N/A | N/A | N/A | N/A | 0.626 | 0.371 | 0.627 | 0.372 | 0.635 | 0.376 | 0.635 | 0.376 | 0.635 | 0.377 |
| | Avg | 0.544 | 0.336 | 0.545 | 0.336 | N/A | N/A | N/A | N/A | N/A | N/A | 0.567 | 0.334 | 0.542 | 0.334 | 0.548 | 0.338 | 0.549 | 0.338 | 0.549 | 0.339 |
| Weather | 96 | 0.158 | 0.209 | 0.162 | 0.217 | 0.159 | 0.218 | 0.159 | 0.210 | 0.158 | 0.216 | 0.160 | 0.209 | 0.160 | 0.209 | 0.160 | 0.209 | 0.160 | 0.209 | 0.161 | 0.207 |
| | 192 | 0.203 | 0.251 | 0.208 | 0.257 | 0.203 | 0.257 | 0.206 | 0.252 | 0.203 | 0.256 | 0.207 | 0.251 | 0.206 | 0.251 | 0.207 | 0.252 | 0.206 | 0.252 | 0.209 | 0.250 |
| | 336 | 0.256 | 0.290 | 0.262 | 0.296 | 0.256 | 0.294 | 0.261 | 0.292 | 0.257 | 0.296 | 0.262 | 0.291 | 0.262 | 0.291 | 0.263 | 0.292 | 0.261 | 0.292 | 0.265 | 0.292 |
| | 720 | 0.338 | 0.343 | 0.341 | 0.346 | 0.333 | 0.344 | 0.340 | 0.343 | 0.334 | 0.346 | 0.340 | 0.342 | 0.342 | 0.343 | 0.344 | 0.344 | 0.340 | 0.344 | 0.348 | 0.345 |
| | Avg | 0.239 | 0.273 | 0.243 | 0.279 | 0.238 | 0.278 | 0.242 | 0.274 | 0.238 | 0.278 | 0.242 | 0.273 | 0.242 | 0.273 | 0.244 | 0.274 | 0.242 | 0.274 | 0.246 | 0.274 |
| Electricity | 96 | 0.156 | 0.248 | 0.157 | 0.247 | N/A | N/A | N/A | N/A | N/A | N/A | 0.156 | 0.247 | 0.156 | 0.247 | 0.156 | 0.247 | 0.156 | 0.248 | 0.156 | 0.247 |
| | 192 | 0.177 | 0.268 | 0.178 | 0.267 | N/A | N/A | N/A | N/A | N/A | N/A | 0.177 | 0.267 | 0.177 | 0.267 | 0.177 | 0.267 | 0.177 | 0.268 | 0.177 | 0.268 |
| | 336 | 0.205 | 0.293 | 0.206 | 0.293 | N/A | N/A | N/A | N/A | N/A | N/A | 0.203 | 0.292 | 0.204 | 0.293 | 0.205 | 0.294 | 0.205 | 0.292 | 0.205 | 0.294 |
| | 720 | 0.270 | 0.346 | 0.271 | 0.346 | N/A | N/A | N/A | N/A | N/A | N/A | 0.267 | 0.343 | 0.269 | 0.344 | 0.271 | 0.346 | 0.270 | 0.345 | 0.271 | 0.346 |
| | Avg | 0.202 | 0.289 | 0.203 | 0.288 | N/A | N/A | N/A | N/A | N/A | N/A | 0.201 | 0.288 | 0.202 | 0.288 | 0.202 | 0.288 | 0.202 | 0.288 | 0.202 | 0.289 |
| Best | | 2 | 2 | 0 | 1 | 1 | 0 | 2 | 3 | 1 | 0 | 1 | 3 | 1 | 2 | 0 | 1 | 0 | 1 | 0 | 0 |
| Second Best | | 3 | 4 | 0 | 1 | 0 | 0 | 0 | 0 | 2 | 0 | 0 | 0 | 1 | 0 | 2 | 0 | 2 | 2 | 1 | 1 |
| **Total** | | 5 | 7 | 0 | 2 | 1 | 0 | 2 | 3 | 3 | 0 | 1 | 3 | 2 | 2 | 2 | 1 | 2 | 3 | 1 | 1 |

Table 5: Raw MSE and MAE results using TimesNet as the backbone forecaster across multiple datasets and horizons. Lower values are better. Red denotes the best value and blue is the second best.

| Dataset | | STD (Ours) MSE | STD (Ours) MAE | MLP MSE | MLP MAE | Logistic MSE | Logistic MAE | RF MSE | RF MAE | XGB MSE | XGB MAE | LSTM MSE | LSTM MAE | GRU MSE | GRU MAE | CNN MSE | CNN MAE | TF. MSE | TF. MAE | TimesNet MSE | TimesNet MAE |
|---|---|---|---|---|---|---|---|---|---|---|---|---|---|---|---|---|---|---|---|---|---|
| ETTh1 | 96 | 0.423 | 0.429 | 0.437 | 0.442 | 0.504 | 0.436 | 0.446 | 0.430 | 0.426 | 0.430 | 0.452 | 0.453 | 0.436 | 0.436 | 0.437 | 0.447 | 0.428 | 0.432 | 0.428 | 0.433 |
| | 192 | 0.451 | 0.448 | 0.470 | 0.459 | 0.533 | 0.458 | 0.477 | 0.452 | 0.458 | 0.452 | 0.490 | 0.474 | 0.461 | 0.455 | 0.473 | 0.470 | 0.464 | 0.454 | 0.467 | 0.458 |
| | 336 | 0.469 | 0.462 | 0.490 | 0.471 | 0.557 | 0.477 | 0.499 | 0.468 | 0.480 | 0.470 | 0.520 | 0.493 | 0.481 | 0.472 | 0.500 | 0.490 | 0.491 | 0.473 | 0.494 | 0.478 |
| | 720 | 0.481 | 0.478 | 0.491 | 0.486 | 0.576 | 0.496 | 0.500 | 0.480 | 0.487 | 0.488 | 0.531 | 0.509 | 0.493 | 0.493 | 0.516 | 0.516 | 0.501 | 0.493 | 0.501 | 0.497 |
| | Avg | 0.456 | 0.454 | 0.472 | 0.465 | 0.543 | 0.466 | 0.480 | 0.458 | 0.463 | 0.455 | 0.498 | 0.482 | 0.468 | 0.464 | 0.482 | 0.476 | 0.471 | 0.463 | 0.472 | 0.465 |
| ETTh2 | 96 | 0.327 | 0.366 | 0.335 | 0.367 | 0.346 | 0.391 | 0.332 | 0.366 | 0.332 | 0.371 | 0.333 | 0.366 | 0.338 | 0.372 | 0.334 | 0.369 | 0.336 | 0.370 | 0.338 | 0.369 |
| | 192 | 0.401 | 0.410 | 0.408 | 0.411 | 0.415 | 0.429 | 0.404 | 0.409 | 0.403 | 0.412 | 0.406 | 0.410 | 0.410 | 0.414 | 0.405 | 0.410 | 0.407 | 0.412 | 0.412 | 0.413 |
| | 336 | 0.433 | 0.440 | 0.443 | 0.441 | 0.443 | 0.453 | 0.439 | 0.439 | 0.437 | 0.441 | 0.443 | 0.442 | 0.442 | 0.443 | 0.438 | 0.439 | 0.441 | 0.442 | 0.447 | 0.443 |
| | 720 | 0.420 | 0.444 | 0.429 | 0.445 | 0.442 | 0.462 | 0.428 | 0.444 | 0.431 | 0.448 | 0.434 | 0.448 | 0.430 | 0.445 | 0.425 | 0.443 | 0.431 | 0.446 | 0.433 | 0.447 |
| | Avg | 0.395 | 0.415 | 0.404 | 0.416 | 0.411 | 0.433 | 0.401 | 0.415 | 0.401 | 0.423 | 0.404 | 0.416 | 0.405 | 0.419 | 0.401 | 0.415 | 0.408 | 0.418 | 0.408 | 0.418 |
| ETTm1 | 96 | 0.403 | 0.417 | 0.417 | 0.417 | 0.417 | 0.414 | 0.420 | 0.416 | 0.415 | 0.417 | 0.411 | 0.416 | 0.415 | 0.418 | 0.412 | 0.414 | 0.412 | 0.414 | 0.421 | 0.419 |
| | 192 | 0.443 | 0.436 | 0.460 | 0.440 | 0.460 | 0.448 | 0.460 | 0.457 | 0.460 | 0.457 | 0.447 | 0.440 | 0.460 | 0.440 | 0.459 | 0.442 | 0.457 | 0.438 | 0.464 | 0.441 |
| | 336 | 0.494 | 0.462 | 0.515 | 0.469 | 0.488 | 0.466 | 0.505 | 0.461 | 0.485 | 0.450 | 0.515 | 0.471 | 0.516 | 0.470 | 0.515 | 0.472 | 0.515 | 0.469 | 0.521 | 0.472 |
| | 720 | 0.592 | 0.508 | 0.617 | 0.517 | 0.557 | 0.508 | 0.632 | 0.464 | 0.534 | 0.474 | 0.625 | 0.520 | 0.620 | 0.518 | 0.621 | 0.522 | 0.623 | 0.519 | 0.625 | 0.520 |
| | Avg | 0.483 | 0.456 | 0.502 | 0.461 | 0.481 | 0.459 | 0.503 | 0.450 | 0.474 | 0.456 | 0.502 | 0.462 | 0.502 | 0.461 | 0.503 | 0.463 | 0.502 | 0.460 | 0.508 | 0.463 |
| ETTm2 | 96 | 0.192 | 0.270 | 0.191 | 0.270 | 0.194 | 0.283 | 0.188 | 0.270 | 0.191 | 0.278 | 0.192 | 0.274 | 0.198 | 0.283 | 0.192 | 0.271 | 0.190 | 0.271 | 0.193 | 0.269 |
| | 192 | 0.258 | 0.309 | 0.255 | 0.310 | 0.255 | 0.318 | 0.248 | 0.308 | 0.253 | 0.316 | 0.255 | 0.313 | 0.261 | 0.319 | 0.256 | 0.310 | 0.254 | 0.311 | 0.259 | 0.310 |
| | 336 | 0.321 | 0.350 | 0.317 | 0.351 | 0.315 | 0.356 | 0.307 | 0.346 | 0.313 | 0.355 | 0.317 | 0.353 | 0.323 | 0.358 | 0.318 | 0.350 | 0.316 | 0.352 | 0.323 | 0.351 |
| | 720 | 0.427 | 0.412 | 0.420 | 0.412 | 0.415 | 0.414 | 0.408 | 0.406 | 0.414 | 0.414 | 0.420 | 0.412 | 0.422 | 0.415 | 0.418 | 0.409 | 0.421 | 0.413 | 0.428 | 0.412 |
| | Avg | 0.300 | 0.335 | 0.296 | 0.335 | 0.325 | 0.342 | 0.313 | 0.333 | 0.318 | 0.341 | 0.322 | 0.339 | 0.326 | 0.344 | 0.321 | 0.335 | 0.320 | 0.337 | 0.301 | 0.336 |
| Traffic | 96 | 0.646 | 0.358 | 0.643 | 0.357 | N/A | N/A | N/A | N/A | N/A | N/A | 0.642 | 0.356 | 0.642 | 0.357 | 0.647 | 0.361 | 0.646 | 0.360 | 0.647 | 0.361 |
| | 192 | 0.650 | 0.366 | 0.654 | 0.366 | N/A | N/A | N/A | N/A | N/A | N/A | 0.652 | 0.365 | 0.652 | 0.366 | 0.659 | 0.371 | 0.654 | 0.367 | 0.659 | 0.371 |
| | 336 | 0.670 | 0.388 | 0.681 | 0.388 | N/A | N/A | N/A | N/A | N/A | N/A | 0.679 | 0.386 | 0.679 | 0.388 | 0.689 | 0.395 | 0.684 | 0.388 | 0.689 | 0.395 |
| | 720 | 0.782 | 0.462 | 0.801 | 0.462 | N/A | N/A | N/A | N/A | N/A | N/A | 0.792 | 0.457 | 0.792 | 0.462 | 0.813 | 0.470 | 0.801 | 0.460 | 0.812 | 0.470 |
| | Avg | 0.687 | 0.394 | 0.720 | 0.418 | N/A | N/A | N/A | N/A | N/A | N/A | 0.691 | 0.391 | 0.691 | 0.393 | 0.702 | 0.414 | 0.733 | 0.417 | 0.702 | 0.399 |
| Weather | 96 | 0.187 | 0.234 | 0.187 | 0.237 | 0.184 | 0.240 | 0.196 | 0.245 | 0.203 | 0.240 | 0.214 | 0.254 | 0.199 | 0.246 | 0.188 | 0.237 | 0.202 | 0.247 | 0.188 | 0.236 |
| | 192 | 0.232 | 0.271 | 0.232 | 0.273 | 0.227 | 0.274 | 0.239 | 0.280 | 0.240 | 0.276 | 0.252 | 0.286 | 0.239 | 0.279 | 0.233 | 0.274 | 0.240 | 0.276 | 0.235 | 0.275 |
| | 336 | 0.284 | 0.308 | 0.283 | 0.310 | 0.275 | 0.307 | 0.289 | 0.315 | 0.281 | 0.310 | 0.295 | 0.318 | 0.286 | 0.314 | 0.285 | 0.310 | 0.282 | 0.310 | 0.289 | 0.312 |
| | 720 | 0.367 | 0.362 | 0.367 | 0.363 | 0.353 | 0.358 | 0.362 | 0.367 | 0.349 | 0.361 | 0.368 | 0.368 | 0.361 | 0.363 | 0.369 | 0.364 | 0.349 | 0.361 | 0.375 | 0.367 |
| | Avg | 0.268 | 0.294 | 0.267 | 0.308 | 0.260 | 0.310 | 0.287 | 0.327 | 0.268 | 0.322 | 0.332 | 0.331 | 0.311 | 0.325 | 0.319 | 0.309 | 0.268 | 0.308 | 0.270 | 0.296 |
| Electricity | 96 | 0.167 | 0.271 | 0.168 | 0.272 | N/A | N/A | N/A | N/A | N/A | N/A | 0.168 | 0.272 | 0.166 | 0.270 | 0.167 | 0.272 | 0.168 | 0.272 | 0.168 | 0.271 |
| | 192 | 0.183 | 0.284 | 0.184 | 0.285 | N/A | N/A | N/A | N/A | N/A | N/A | 0.184 | 0.285 | 0.182 | 0.284 | 0.183 | 0.285 | 0.184 | 0.285 | 0.184 | 0.285 |
| | 336 | 0.202 | 0.303 | 0.204 | 0.304 | N/A | N/A | N/A | N/A | N/A | N/A | 0.204 | 0.304 | 0.201 | 0.303 | 0.203 | 0.304 | 0.204 | 0.304 | 0.203 | 0.304 |
| | 720 | 0.254 | 0.344 | 0.257 | 0.347 | N/A | N/A | N/A | N/A | N/A | N/A | 0.257 | 0.347 | 0.252 | 0.343 | 0.256 | 0.346 | 0.257 | 0.347 | 0.256 | 0.347 |
| | Avg | 0.202 | 0.301 | 0.203 | 0.302 | N/A | N/A | N/A | N/A | N/A | N/A | 0.203 | 0.302 | 0.201 | 0.300 | 0.202 | 0.301 | 0.203 | 0.302 | 0.203 | 0.302 |
| Best | | 3 | 3 | 1 | 0 | 1 | 0 | 0 | 3 | 1 | 0 | 0 | 1 | 1 | 1 | 0 | 1 | 0 | 0 | 0 | 0 |
| Second Best | | 2 | 3 | 1 | 2 | 1 | 0 | 1 | 0 | 2 | 2 | 1 | 1 | 1 | 1 | 1 | 1 | 0 | 0 | 0 | 01 |
| Total | | 5 | 6 | 2 | 2 | 2 | 0 | 1 | 3 | 3 | 2 | 1 | 2 | 2 | 2 | 1 | 2 | 0 | 0 | 0 | 1 |

Table 6: Raw MSE and MAE results using TimeXer as the backbone forecaster across multiple datasets and horizons. Lower values are better. Red denotes the best value and blue is the second best.

| Dataset | | STD (Ours) MSE | STD (Ours) MAE | MLP MSE | MLP MAE | Logistic MSE | Logistic MAE | RF MSE | RF MAE | XGB MSE | XGB MAE | LSTM MSE | LSTM MAE | GRU MSE | GRU MAE | CNN MSE | CNN MAE | TF. MSE | TF. MAE | TimeXer MSE | TimeXer MAE |
|---|---|---|---|---|---|---|---|---|---|---|---|---|---|---|---|---|---|---|---|---|---|
| ETTh1 | 96 | 0.394 | 0.418 | 0.397 | 0.407 | 0.492 | 0.417 | 0.401 | 0.409 | 0.408 | 0.410 | 0.405 | 0.419 | 0.431 | 0.421 | 0.400 | 0.415 | 0.399 | 0.417 | 0.395 | 0.407 |
| | 192 | 0.441 | 0.449 | 0.447 | 0.438 | 0.534 | 0.449 | 0.449 | 0.442 | 0.456 | 0.443 | 0.452 | 0.450 | 0.487 | 0.455 | 0.447 | 0.447 | 0.445 | 0.446 | 0.447 | 0.441 |
| | 336 | 0.489 | 0.481 | 0.494 | 0.469 | 0.583 | 0.483 | 0.497 | 0.474 | 0.504 | 0.475 | 0.502 | 0.482 | 0.543 | 0.488 | 0.501 | 0.482 | 0.493 | 0.476 | 0.500 | 0.474 |
| | 720 | 0.556 | 0.533 | 0.543 | 0.519 | 0.662 | 0.535 | 0.548 | 0.520 | 0.561 | 0.523 | 0.560 | 0.531 | 0.628 | 0.546 | 0.573 | 0.538 | 0.554 | 0.524 | 0.557 | 0.524 |
| | Avg | 0.470 | 0.470 | 0.470 | 0.458 | 0.568 | 0.471 | 0.474 | 0.461 | 0.482 | 0.463 | 0.480 | 0.470 | 0.522 | 0.478 | 0.480 | 0.468 | 0.470 | 0.466 | 0.475 | 0.462 |
| ETTh2 | 96 | 0.290 | 0.343 | 0.294 | 0.346 | 0.324 | 0.397 | 0.291 | 0.346 | 0.293 | 0.350 | 0.293 | 0.345 | 0.292 | 0.345 | 0.292 | 0.343 | 0.292 | 0.344 | 0.293 | 0.344 |
| | 192 | 0.374 | 0.394 | 0.381 | 0.399 | 0.405 | 0.443 | 0.375 | 0.397 | 0.378 | 0.402 | 0.380 | 0.399 | 0.377 | 0.397 | 0.378 | 0.396 | 0.375 | 0.395 | 0.379 | 0.397 |
| | 336 | 0.421 | 0.433 | 0.430 | 0.439 | 0.447 | 0.475 | 0.423 | 0.435 | 0.428 | 0.441 | 0.430 | 0.437 | 0.426 | 0.436 | 0.426 | 0.435 | 0.421 | 0.434 | 0.428 | 0.436 |
| | 720 | 0.439 | 0.453 | 0.449 | 0.459 | 0.482 | 0.499 | 0.441 | 0.455 | 0.451 | 0.464 | 0.446 | 0.457 | 0.445 | 0.457 | 0.442 | 0.454 | 0.438 | 0.454 | 0.445 | 0.456 |
| | Avg | 0.381 | 0.406 | 0.388 | 0.411 | 0.414 | 0.454 | 0.383 | 0.408 | 0.387 | 0.414 | 0.387 | 0.409 | 0.385 | 0.409 | 0.384 | 0.407 | 0.381 | 0.407 | 0.386 | 0.408 |
| ETTm1 | 96 | 0.313 | 0.357 | 0.319 | 0.360 | 0.314 | 0.357 | 0.318 | 0.359 | 0.316 | 0.361 | 0.321 | 0.360 | 0.321 | 0.360 | 0.320 | 0.361 | 0.320 | 0.360 | 0.322 | 0.361 |
| | 192 | 0.367 | 0.391 | 0.382 | 0.399 | 0.375 | 0.395 | 0.380 | 0.397 | 0.378 | 0.399 | 0.385 | 0.399 | 0.384 | 0.399 | 0.383 | 0.400 | 0.383 | 0.399 | 0.385 | 0.400 |
| | 336 | 0.421 | 0.425 | 0.445 | 0.437 | 0.436 | 0.433 | 0.442 | 0.435 | 0.445 | 0.438 | 0.448 | 0.437 | 0.446 | 0.437 | 0.446 | 0.438 | 0.445 | 0.436 | 0.449 | 0.438 |
| | 720 | 0.524 | 0.481 | 0.558 | 0.496 | 0.547 | 0.491 | 0.554 | 0.493 | 0.559 | 0.494 | 0.562 | 0.496 | 0.560 | 0.495 | 0.560 | 0.497 | 0.559 | 0.494 | 0.563 | 0.497 |
| | Avg | 0.406 | 0.414 | 0.426 | 0.423 | 0.418 | 0.419 | 0.424 | 0.421 | 0.424 | 0.423 | 0.429 | 0.423 | 0.428 | 0.423 | 0.427 | 0.424 | 0.427 | 0.422 | 0.411 | 0.424 |
| ETTm2 | 96 | 0.169 | 0.267 | 0.172 | 0.259 | 0.171 | 0.266 | 0.170 | 0.258 | 0.172 | 0.265 | 0.385 | 0.407 | 0.173 | 0.261 | 0.181 | 0.271 | 0.191 | 0.262 | 0.174 | 0.259 |
| | 192 | 0.232 | 0.308 | 0.237 | 0.303 | 0.233 | 0.306 | 0.232 | 0.301 | 0.235 | 0.307 | 0.434 | 0.438 | 0.239 | 0.304 | 0.247 | 0.315 | 0.251 | 0.305 | 0.241 | 0.304 |
| | 336 | 0.299 | 0.349 | 0.304 | 0.347 | 0.299 | 0.347 | 0.298 | 0.343 | 0.300 | 0.348 | 0.487 | 0.469 | 0.307 | 0.347 | 0.314 | 0.357 | 0.308 | 0.347 | 0.311 | 0.348 |
| | 720 | 0.408 | 0.410 | 0.410 | 0.410 | 0.401 | 0.407 | 0.403 | 0.405 | 0.404 | 0.409 | 0.585 | 0.519 | 0.414 | 0.409 | 0.414 | 0.415 | 0.406 | 0.408 | 0.421 | 0.411 |
| | Avg | 0.277 | 0.334 | 0.281 | 0.330 | 0.276 | 0.332 | 0.276 | 0.327 | 0.278 | 0.332 | 0.473 | 0.458 | 0.283 | 0.330 | 0.289 | 0.339 | 0.287 | 0.331 | 0.287 | 0.331 |
| Traffic | 96 | 0.468 | 0.301 | 0.469 | 0.300 | N/A | N/A | N/A | N/A | N/A | N/A | 0.467 | 0.298 | 0.468 | 0.298 | 0.471 | 0.299 | 0.471 | 0.300 | 0.471 | 0.303 |
| | 192 | 0.471 | 0.302 | 0.471 | 0.300 | N/A | N/A | N/A | N/A | N/A | N/A | 0.469 | 0.298 | 0.470 | 0.299 | 0.473 | 0.299 | 0.473 | 0.300 | 0.473 | 0.303 |
| | 336 | 0.470 | 0.300 | 0.470 | 0.298 | N/A | N/A | N/A | N/A | N/A | N/A | 0.468 | 0.296 | 0.469 | 0.297 | 0.473 | 0.298 | 0.473 | 0.298 | 0.473 | 0.301 |
| | 720 | 0.476 | 0.302 | 0.477 | 0.300 | N/A | N/A | N/A | N/A | N/A | N/A | 0.475 | 0.298 | 0.475 | 0.299 | 0.479 | 0.300 | 0.479 | 0.301 | 0.479 | 0.303 |
| | Avg | 0.471 | 0.301 | 0.472 | 0.300 | N/A | N/A | N/A | N/A | N/A | N/A | 0.470 | 0.298 | 0.471 | 0.298 | 0.474 | 0.299 | 0.474 | 0.300 | 0.474 | 0.303 |
| Weather | 96 | 0.159 | 0.207 | 0.162 | 0.217 | 0.159 | 0.218 | 0.159 | 0.210 | 0.158 | 0.216 | 0.160 | 0.209 | 0.160 | 0.209 | 0.160 | 0.209 | 0.160 | 0.209 | 0.161 | 0.207 |
| | 192 | 0.205 | 0.248 | 0.208 | 0.257 | 0.203 | 0.257 | 0.206 | 0.252 | 0.203 | 0.256 | 0.207 | 0.251 | 0.206 | 0.251 | 0.207 | 0.252 | 0.206 | 0.252 | 0.209 | 0.250 |
| | 336 | 0.260 | 0.289 | 0.262 | 0.296 | 0.256 | 0.294 | 0.261 | 0.292 | 0.257 | 0.296 | 0.262 | 0.291 | 0.262 | 0.291 | 0.263 | 0.292 | 0.261 | 0.292 | 0.265 | 0.292 |
| | 720 | 0.338 | 0.340 | 0.341 | 0.346 | 0.333 | 0.344 | 0.340 | 0.343 | 0.334 | 0.346 | 0.340 | 0.342 | 0.342 | 0.343 | 0.344 | 0.344 | 0.340 | 0.344 | 0.348 | 0.345 |
| | Avg | 0.241 | 0.271 | 0.243 | 0.279 | 0.238 | 0.278 | 0.242 | 0.274 | 0.238 | 0.279 | 0.242 | 0.273 | 0.242 | 0.273 | 0.243 | 0.274 | 0.242 | 0.274 | 0.246 | 0.274 |
| Electricity | 96 | 0.139 | 0.240 | 0.140 | 0.241 | N/A | N/A | N/A | N/A | N/A | N/A | 0.139 | 0.240 | 0.140 | 0.241 | 0.140 | 0.241 | 0.139 | 0.239 | 0.140 | 0.242 |
| | 192 | 0.165 | 0.266 | 0.167 | 0.271 | N/A | N/A | N/A | N/A | N/A | N/A | 0.166 | 0.269 | 0.167 | 0.270 | 0.167 | 0.270 | 0.164 | 0.266 | 0.167 | 0.271 |
| | 336 | 0.200 | 0.303 | 0.205 | 0.310 | N/A | N/A | N/A | N/A | N/A | N/A | 0.202 | 0.307 | 0.204 | 0.309 | 0.205 | 0.309 | 0.199 | 0.303 | 0.204 | 0.311 |
| | 720 | 0.294 | 0.385 | 0.304 | 0.394 | N/A | N/A | N/A | N/A | N/A | N/A | 0.298 | 0.390 | 0.303 | 0.394 | 0.304 | 0.394 | 0.294 | 0.385 | 0.304 | 0.395 |
| | Avg | 0.200 | 0.298 | 0.204 | 0.304 | N/A | N/A | N/A | N/A | N/A | N/A | 0.201 | 0.302 | 0.204 | 0.304 | 0.204 | 0.304 | 0.199 | 0.298 | 0.203 | 0.302 |
| Best | | 3 | 4 | 1 | 1 | 2 | 0 | 1 | 1 | 1 | 0 | 1 | 1 | 0 | 1 | 0 | 0 | 2 | 1 | 0 | 0 |
| Second Best | | 4 | 0 | 0 | 1 | 0 | 1 | 1 | 1 | 0 | 0 | 0 | 2 | 1 | 2 | 0 | 2 | 1 | 1 | 1 | 1 |
| Total | | 7 | 4 | 1 | 2 | 2 | 1 | 2 | 2 | 1 | 0 | 1 | 3 | 1 | 3 | 0 | 2 | 3 | 2 | 1 | 1 |

Table 7: Raw MSE and MAE results using TimeBridge as the backbone forecaster across multiple datasets and horizons. Lower values are better. Red denotes the best value and blue is the second best.

| Dataset | | STD (Ours) MSE | MAE | MLP MSE | MAE | Logistic MSE | MAE | RF MSE | MAE | XGB MSE | MAE | LSTM MSE | MAE | GRU MSE | MAE | CNN MSE | MAE | TF. MSE | MAE | TimeBridge MSE | MAE |
|---|---|---|---|---|---|---|---|---|---|---|---|---|---|---|---|---|---|---|---|---|---|
| ETTh1 | 96 | 0.382 | 0.404 | 0.385 | 0.402 | 0.388 | 0.405 | 0.388 | 0.402 | 0.388 | 0.416 | 0.435 | 0.447 | 0.387 | 0.413 | 0.390 | 0.410 | 0.389 | 0.413 | 0.385 | 0.401 |
| | 192 | 0.429 | 0.433 | 0.432 | 0.429 | 0.435 | 0.434 | 0.435 | 0.430 | 0.435 | 0.443 | 0.487 | 0.473 | 0.435 | 0.440 | 0.438 | 0.440 | 0.436 | 0.439 | 0.434 | 0.431 |
| | 336 | 0.469 | 0.456 | 0.471 | 0.451 | 0.477 | 0.458 | 0.476 | 0.452 | 0.475 | 0.465 | 0.536 | 0.498 | 0.479 | 0.466 | 0.481 | 0.465 | 0.479 | 0.462 | 0.478 | 0.456 |
| | 720 | 0.495 | 0.485 | 0.492 | 0.481 | 0.508 | 0.491 | 0.492 | 0.478 | 0.496 | 0.490 | 0.548 | 0.520 | 0.514 | 0.501 | 0.510 | 0.502 | 0.506 | 0.494 | 0.499 | 0.487 |
| | Avg | 0.444 | 0.444 | 0.445 | 0.441 | 0.452 | 0.447 | 0.448 | 0.441 | 0.449 | 0.454 | 0.502 | 0.485 | 0.454 | 0.455 | 0.455 | 0.454 | 0.453 | 0.452 | 0.449 | 0.444 |
| ETTh2 | 96 | 0.296 | 0.346 | 0.301 | 0.350 | 0.322 | 0.395 | 0.298 | 0.351 | 0.300 | 0.354 | 0.298 | 0.348 | 0.297 | 0.348 | 0.298 | 0.349 | 0.296 | 0.351 | 0.300 | 0.348 |
| | 192 | 0.379 | 0.396 | 0.385 | 0.401 | 0.400 | 0.440 | 0.380 | 0.400 | 0.381 | 0.404 | 0.380 | 0.399 | 0.380 | 0.398 | 0.378 | 0.398 | 0.375 | 0.400 | 0.383 | 0.399 |
| | 336 | 0.431 | 0.436 | 0.437 | 0.441 | 0.446 | 0.473 | 0.431 | 0.438 | 0.432 | 0.442 | 0.430 | 0.438 | 0.432 | 0.438 | 0.428 | 0.437 | 0.425 | 0.438 | 0.435 | 0.439 |
| | 720 | 0.444 | 0.454 | 0.454 | 0.461 | 0.478 | 0.495 | 0.444 | 0.456 | 0.453 | 0.463 | 0.444 | 0.456 | 0.449 | 0.458 | 0.444 | 0.455 | 0.446 | 0.458 | 0.448 | 0.456 |
| | Avg | 0.388 | 0.408 | 0.394 | 0.413 | 0.411 | 0.451 | 0.388 | 0.411 | 0.391 | 0.416 | 0.388 | 0.410 | 0.390 | 0.411 | 0.387 | 0.410 | 0.386 | 0.412 | 0.392 | 0.411 |
| ETTm1 | 96 | 0.320 | 0.360 | 0.322 | 0.363 | 0.320 | 0.361 | 0.324 | 0.362 | 0.320 | 0.362 | 0.326 | 0.371 | 0.326 | 0.364 | 0.333 | 0.374 | 0.324 | 0.362 | 0.325 | 0.363 |
| | 192 | 0.375 | 0.394 | 0.377 | 0.397 | 0.375 | 0.395 | 0.379 | 0.397 | 0.374 | 0.395 | 0.383 | 0.398 | 0.383 | 0.399 | 0.392 | 0.409 | 0.381 | 0.397 | 0.382 | 0.398 |
| | 336 | 0.424 | 0.426 | 0.427 | 0.427 | 0.424 | 0.426 | 0.429 | 0.428 | 0.422 | 0.426 | 0.432 | 0.435 | 0.433 | 0.430 | 0.450 | 0.442 | 0.432 | 0.429 | 0.433 | 0.430 |
| | 720 | 0.521 | 0.474 | 0.523 | 0.474 | 0.519 | 0.473 | 0.527 | 0.476 | 0.516 | 0.472 | 0.531 | 0.478 | 0.531 | 0.478 | 0.558 | 0.490 | 0.530 | 0.477 | 0.532 | 0.478 |
| | Avg | 0.410 | 0.414 | 0.412 | 0.415 | 0.409 | 0.414 | 0.415 | 0.416 | 0.408 | 0.414 | 0.418 | 0.420 | 0.419 | 0.418 | 0.433 | 0.429 | 0.417 | 0.416 | 0.418 | 0.417 |
| ETTm2 | 96 | 0.179 | 0.260 | 0.180 | 0.263 | 0.180 | 0.271 | 0.177 | 0.283 | 0.180 | 0.270 | 0.183 | 0.268 | 0.183 | 0.267 | 0.180 | 0.262 | 0.184 | 0.338 | 0.181 | 0.262 |
| | 192 | 0.245 | 0.304 | 0.246 | 0.306 | 0.241 | 0.310 | 0.239 | 0.319 | 0.241 | 0.310 | 0.247 | 0.310 | 0.247 | 0.309 | 0.246 | 0.305 | 0.246 | 0.373 | 0.248 | 0.305 |
| | 336 | 0.311 | 0.346 | 0.314 | 0.347 | 0.303 | 0.349 | 0.300 | 0.353 | 0.301 | 0.349 | 0.312 | 0.350 | 0.311 | 0.349 | 0.311 | 0.346 | 0.308 | 0.406 | 0.398 | 0.409 |
| | 720 | 0.418 | 0.407 | 0.419 | 0.408 | 0.404 | 0.407 | 0.403 | 0.408 | 0.404 | 0.409 | 0.416 | 0.409 | 0.417 | 0.409 | 0.418 | 0.407 | 0.414 | 0.459 | 0.422 | 0.408 |
| | Avg | 0.288 | 0.329 | 0.290 | 0.331 | 0.282 | 0.334 | 0.280 | 0.341 | 0.282 | 0.335 | 0.289 | 0.335 | 0.289 | 0.334 | 0.289 | 0.330 | 0.288 | 0.394 | 0.312 | 0.346 |
| Traffic | 96 | 0.534 | 0.358 | 0.553 | 0.369 | N/A | N/A | N/A | N/A | N/A | N/A | 0.544 | 0.362 | 0.545 | 0.362 | 0.553 | 0.369 | 0.553 | 0.369 | 0.553 | 0.369 |
| | 192 | 0.579 | 0.375 | 0.595 | 0.385 | N/A | N/A | N/A | N/A | N/A | N/A | 0.584 | 0.377 | 0.584 | 0.377 | 0.595 | 0.385 | 0.595 | 0.385 | 0.594 | 0.385 |
| | 336 | 0.654 | 0.397 | 0.664 | 0.405 | N/A | N/A | N/A | N/A | N/A | N/A | 0.650 | 0.396 | 0.651 | 0.397 | 0.664 | 0.405 | 0.664 | 0.405 | 0.664 | 0.405 |
| | 720 | 0.805 | 0.450 | 0.813 | 0.456 | N/A | N/A | N/A | N/A | N/A | N/A | 0.796 | 0.447 | 0.798 | 0.448 | 0.813 | 0.456 | 0.796 | 0.450 | 0.811 | 0.456 |
| | Avg | 0.643 | 0.395 | 0.656 | 0.404 | N/A | N/A | N/A | N/A | N/A | N/A | 0.644 | 0.396 | 0.645 | 0.396 | 0.656 | 0.404 | 0.652 | 0.402 | 0.656 | 0.404 |
| Weather | 96 | 0.159 | 0.208 | 0.160 | 0.211 | 0.160 | 0.220 | 0.161 | 0.211 | 0.160 | 0.219 | 0.160 | 0.210 | 0.161 | 0.210 | 0.161 | 0.209 | 0.160 | 0.210 | 0.162 | 0.209 |
| | 192 | 0.205 | 0.249 | 0.203 | 0.250 | 0.203 | 0.257 | 0.205 | 0.252 | 0.211 | 0.271 | 0.205 | 0.251 | 0.207 | 0.251 | 0.206 | 0.250 | 0.204 | 0.251 | 0.208 | 0.250 |
| | 336 | 0.259 | 0.290 | 0.256 | 0.288 | 0.254 | 0.294 | 0.259 | 0.291 | 0.262 | 0.307 | 0.259 | 0.290 | 0.262 | 0.291 | 0.261 | 0.290 | 0.257 | 0.291 | 0.263 | 0.291 |
| | 720 | 0.340 | 0.342 | 0.336 | 0.338 | 0.330 | 0.341 | 0.337 | 0.341 | 0.335 | 0.353 | 0.337 | 0.340 | 0.341 | 0.342 | 0.340 | 0.340 | 0.335 | 0.340 | 0.345 | 0.343 |
| | Avg | 0.241 | 0.272 | 0.239 | 0.272 | 0.237 | 0.278 | 0.241 | 0.274 | 0.242 | 0.288 | 0.240 | 0.273 | 0.243 | 0.274 | 0.242 | 0.275 | 0.239 | 0.273 | 0.245 | 0.273 |
| Electricity | 96 | 0.195 | 0.282 | 0.195 | 0.281 | N/A | N/A | N/A | N/A | N/A | N/A | 0.194 | 0.280 | 0.195 | 0.281 | 0.195 | 0.282 | 0.194 | 0.280 | 0.195 | 0.282 |
| | 192 | 0.214 | 0.299 | 0.214 | 0.299 | N/A | N/A | N/A | N/A | N/A | N/A | 0.213 | 0.297 | 0.214 | 0.298 | 0.214 | 0.299 | 0.213 | 0.298 | 0.214 | 0.299 |
| | 336 | 0.240 | 0.322 | 0.240 | 0.322 | N/A | N/A | N/A | N/A | N/A | N/A | 0.238 | 0.320 | 0.239 | 0.321 | 0.240 | 0.322 | 0.238 | 0.320 | 0.240 | 0.322 |
| | 720 | 0.296 | 0.368 | 0.295 | 0.366 | N/A | N/A | N/A | N/A | N/A | N/A | 0.293 | 0.363 | 0.295 | 0.365 | 0.296 | 0.366 | 0.294 | 0.364 | 0.295 | 0.366 |
| | Avg | 0.236 | 0.318 | 0.236 | 0.317 | N/A | N/A | N/A | N/A | N/A | N/A | 0.235 | 0.315 | 0.236 | 0.316 | 0.236 | 0.317 | 0.235 | 0.316 | 0.236 | 0.317 |
| Best | | 2 | 5 | 0 | 2 | 1 | 1 | 1 | 1 | 1 | 1 | 1 | 1 | 0 | 0 | 0 | 0 | 2 | 0 | 0 | 0 |
| Second Best | | 1 | 1 | 3 | 1 | 2 | 0 | 0 | 0 | 0 | 0 | 1 | 3 | 1 | 2 | 2 | 2 | 0 | 2 | 1 | 2 |
| Total | | 3 | 6 | 3 | 3 | 3 | 1 | 1 | 1 | 1 | 1 | 2 | 4 | 1 | 2 | 2 | 2 | 2 | 2 | 1 | 2 |

Table 8: Average Error Reduction in MSE compared to TimeBridge for different UEC methods (the lower the better, negative means improvement). N/A indicates that the method failed to converge or crashed during training. Bold and underline denote best and second-best results, respectively.

| Method | ETTh1 | ETTh2 | ETTm1 | ETTm2 | Traffic | Weather | Electricity |
|---|---|---|---|---|---|---|---|
| AR (No Correction) | 0.00 | 0.00 | 0.00 | 0.00 | 0.00 | 0.00 | 0.00 |
| UEC-MLP | -0.84 | 0.69 | -1.35 | -7.24 | 0.09 | -2.26 | -0.03 |
| UEC-Logistic | 0.67 | 5.08 | -2.06 | -9.65 | N/A | **-3.07** | N/A |
| UEC-Random Forest | -0.21 | -0.84 | -0.83 | **-10.4** | N/A | -1.61 | N/A |
| UEC-XGBoost | -0.10 | -0.06 | **-2.41** | -9.81 | N/A | -0.87 | N/A |
| UEC-LSTM | 11.6 | -0.86 | -0.01 | -7.33 | -1.82 | -1.60 | **-0.64** |
| UEC-GRU | 1.08 | -0.51 | 0.10 | -7.29 | -1.65 | -0.62 | -0.21 |
| UEC-CNN | 1.31 | -1.17 | 3.64 | -7.57 | 0.09 | -0.92 | 0.05 |
| UEC-Transformer | 0.80 | **-1.49** | -0.28 | -7.78 | -0.54 | -2.19 | -0.61 |
| UEC-STD | **-1.15** | -1.03 | -1.92 | -7.63 | **-1.91** | -1.43 | 0.02 |

Table 9: Average Error Reduction in MAE compared to TimeBridge for different UEC methods (the lower the better, negative means improvement). N/A indicates that the method failed to converge or crashed during training. Bold and underline denote best and second-best results, respectively.

| Method | ETTh1 | ETTh2 | ETTm1 | ETTm2 | Traffic | Weather | Electricity |
|---|---|---|---|---|---|---|---|
| AR (No Correction) | 0.00 | 0.00 | 0.00 | 0.00 | 0.00 | 0.00 | 0.00 |
| UEC-MLP | -0.77 | 0.66 | -0.57 | -4.36 | -0.01 | **-0.40** | -0.09 |
| UEC-Logistic | 0.70 | 9.76 | -0.90 | -3.29 | N/A | 1.77 | N/A |
| UEC-Random Forest | **-0.81** | 0.21 | -0.41 | -1.54 | N/A | 0.25 | N/A |
| UEC-XGBoost | 2.19 | 1.26 | -0.83 | -3.25 | N/A | 5.39 | N/A |
| UEC-LSTM | 9.15 | -0.07 | 0.73 | -3.27 | -1.98 | -0.21 | **-0.60** |
| UEC-GRU | 2.49 | 0.03 | 0.05 | -3.50 | -1.86 | 0.18 | -0.20 |
| UEC-CNN | 2.32 | -0.21 | 2.73 | -4.58 | -0.01 | -0.21 | 0.01 |
| UEC-Transformer | 1.85 | 0.32 | -0.26 | 13.9 | -0.36 | 0.02 | -0.52 |
| UEC-STD | 0.13 | **-0.61** | **-0.92** | **-4.82** | **-2.13** | **-0.40** | -0.08 |

## C.4 HYPERPARAMETERS

### C.4.1 HYPERPARAMETERS OF BACKBONES

The hyperparameters for the backbone models (TimeMixer, TimesNet, and TimeXer) are adopted directly from the official Time-Series-Library repository by THUML [2], in line with their experimental settings. These settings (such as look-back length, model depth, hidden sizes, and other architecture-specific parameters) are consistent with those used in the TSLib implementation. At the same time, some hyperparameters are dataset-dependent, meaning that choices like sequence length, batch size, or certain regularization parameters vary depending on the particular dataset in use.

### C.4.2 HYPERPARAMETERS OF UEC

All UEC models in our experiments were trained using the same set of hyperparameters summarized in Table 10. The same set of corrections was constructed from the validation split $\mathcal{D}_{\text{val}}$, with a 70/30 split for training and early stopping / $\beta$ tuning, was used for all UEC models. The correction strength $\beta$ was selected separately for MSE and MAE using a balanced validation strategy, and it is reported in Table 11. Based on the results in Table 14, we chose the Huber loss to train all UEC models, as it consistently led to the best performance across both MSE and MAE metrics.

---

[2]https://github.com/thuml/Time-Series-Library

Table 10: Default Training Parameters of UEC

| Parameter | Value / Description |
|---|---|
| Correction data | $\mathcal{U}_{\text{train}}$ / $\mathcal{U}_{\text{val}}$ (70%/30%) from $\mathcal{D}_{\text{val}}$ |
| Training procedure | Follows Sect. 2.2 and Sect. 2.3 |
| Number of training steps | 100 |
| Batch size | 64 |
| Loss | Huber (HL) Loss |
| Correction strength $\beta$ | Selected separately for MSE and MAE refer to Table 11 |

## C.5 DETAILS ON MODEL ANALYSIS

Table 13 compares the averaged MSE and MAE of direct forecasting (DF) and autoregressive (AR) methods across models, showing that AR consistently outperforms DF.

Figure 4 provides qualitative examples on the TRAFFIC dataset, illustrating how UEC mitigates collapse by restoring variance and correcting drift.

Table 14 presents the impact of different training losses on UEC performance for ETTh1, indicating Huber loss often yields the best results.

Figure 6 demonstrates performance improvements of UEC-enhanced backbones across multiple prediction lengths, highlighting consistent gains over standard backbone predictions.

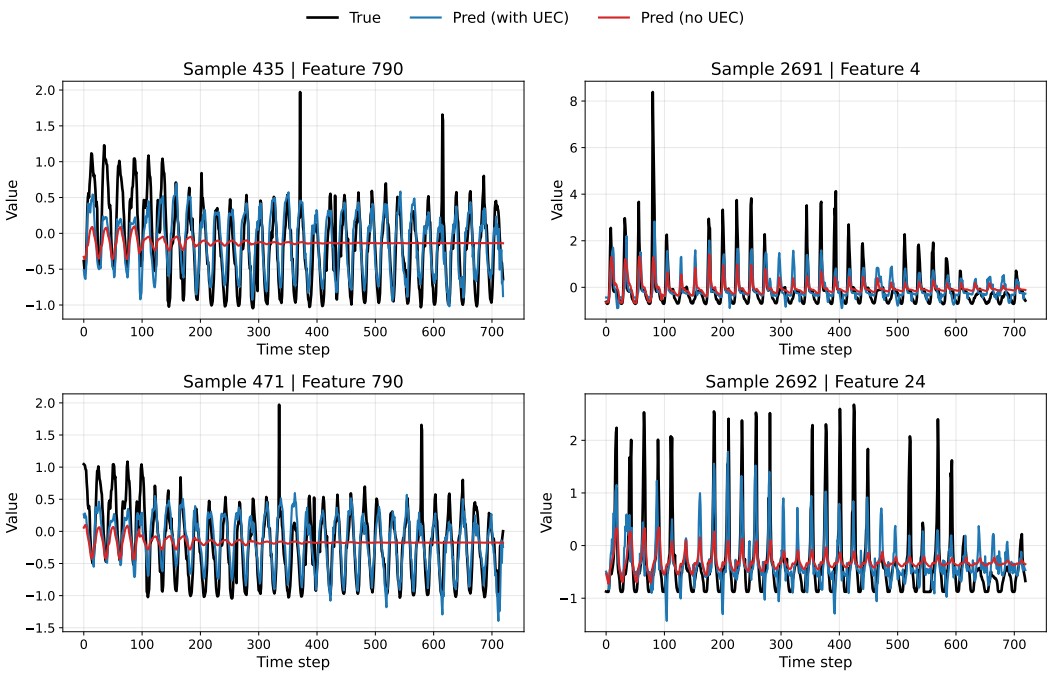

Figure 4: Qualitative examples on TRAFFIC using TimesNet as backbone model (prediction length = 720). Each panel shows the ground truth, prediction with UEC, and prediction without UEC. UEC mitigates collapse by restoring variance and correcting drift.

## C.6 KERNEL SIZE SENSITIVITY ANALYSIS

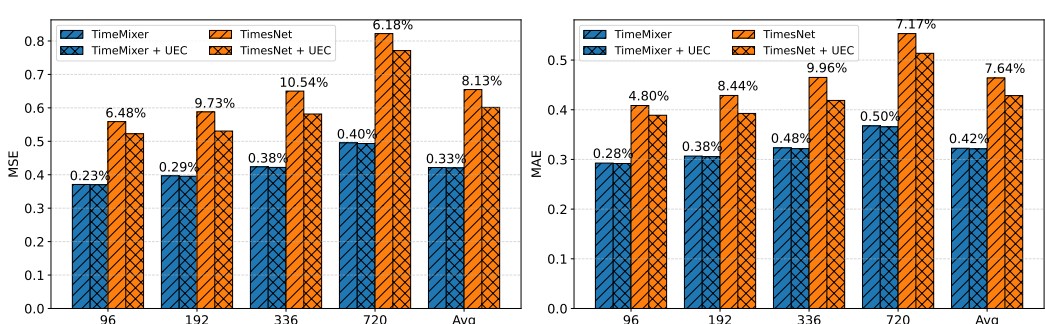

Figure 5: Performance of extended training across different prediction lengths: 96, 192, 336, and 720. Backbone models (TimeMixer and TimesNet) are compared with their corresponding UEC-enhanced versions. % improvement is annotated on top of each bar pair.

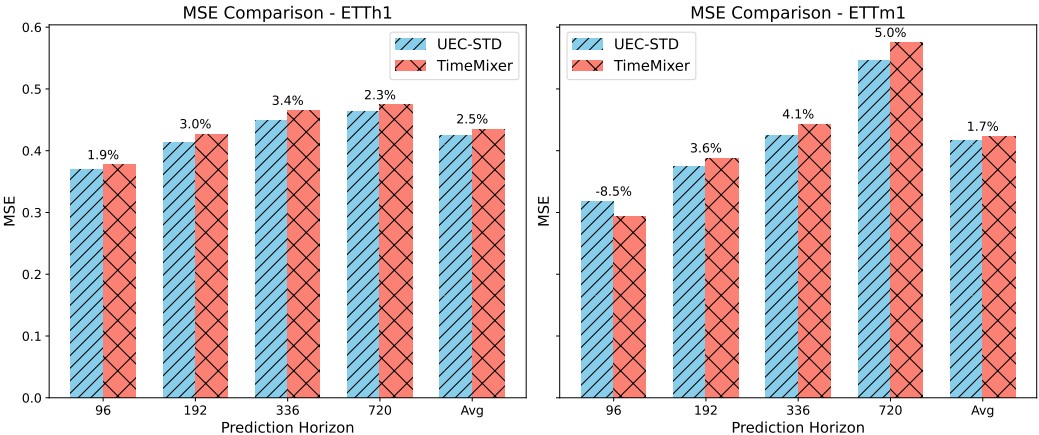

Figure 6: Performance across different prediction lengths: 96, 192, 336, and 720. TimeMixer is compared with its corresponding UEC-enhanced versions on ETTh1 and ETTm1 datasets. % improvement is annotated on top of each bar pair.

Table 11: Found Correction Strength $\beta$ for UEC Models Across Datasets and Backbones

| Dataset | Backbone | STD (Ours) | | MLP | | Logistic | | RF | | XGB | | LSTM | | GRU | | CNN | | TF. | |
|---|---|---|---|---|---|---|---|---|---|---|---|---|---|---|---|---|---|---|---|
| | | MSE | MAE | MSE | MAE | MSE | MAE | MSE | MAE | MSE | MAE | MSE | MAE | MSE | MAE | MSE | MAE | MSE | MAE |
| ETTh1 | TimeMixer | 0.3 | 0.3 | 0.7 | 0.5 | 0.1 | 0.1 | 0.7 | 0.3 | 0.3 | 0.1 | 0.3 | 0.3 | 0.3 | 0.5 | 0.3 | 0.5 | 0.3 | 0.3 |
| | TimesNet | 0.5 | 0.3 | 0.5 | 0.3 | 0.3 | 0.3 | 0.7 | 0.3 | 0.3 | 0.1 | 0.3 | 0.3 | 0.3 | 0.5 | 0.3 | 0.1 | 0.3 | 0.5 |
| | TimeXer | 0.3 | 0.3 | 0.1 | 0.3 | 0.3 | 0.3 | 0.5 | 0.1 | 0.3 | 0.1 | 0.3 | 0.3 | 0.3 | 0.5 | 0.3 | 0.1 | 0.3 | 0.3 |
| | TimeBridge | 0.3 | 0.3 | 0.3 | 0.3 | 0.1 | 0.1 | 0.5 | 0.3 | 0.3 | 0.3 | 0.3 | 0.3 | 0.3 | 0.5 | 0.3 | 0.3 | 0.3 | 0.3 |
| ETTh2 | TimeMixer | 0.1 | 0.1 | 0.1 | 0.1 | 0.1 | 0.1 | 0.1 | 0.1 | 0.1 | 0.1 | 0.1 | 0.1 | 0.1 | 0.1 | 0.1 | 0.1 | 0.1 | 0.1 |
| | TimesNet | 0.1 | 0.1 | 0.1 | 0.1 | 0.1 | 0.1 | 0.1 | 0.1 | 0.1 | 0.1 | 0.1 | 0.1 | 0.1 | 0.1 | 0.1 | 0.1 | 0.1 | 0.1 |
| | TimeXer | 0.1 | 0.1 | 0.1 | 0.1 | 0.1 | 0.1 | 0.1 | 0.1 | 0.1 | 0.1 | 0.1 | 0.1 | 0.1 | 0.1 | 0.1 | 0.1 | 0.3 | 0.1 |
| | TimeBridge | 0.1 | 0.1 | 0.1 | 0.1 | 0.1 | 0.1 | 0.1 | 0.1 | 0.1 | 0.1 | 0.1 | 0.1 | 0.1 | 0.1 | 0.1 | 0.1 | 0.1 | 0.1 |
| ETTm1 | TimeMixer | 0.3 | 0.5 | 0.1 | 0.3 | 0.1 | 0.1 | 0.1 | 0.1 | 0.1 | 0.1 | 0.1 | 0.3 | 0.1 | 0.3 | 0.1 | 0.3 | 0.1 | 0.1 |
| | TimesNet | 0.3 | 0.5 | 0.1 | 0.1 | 0.3 | 0.1 | 0.1 | 0.1 | 0.1 | 0.1 | 0.1 | 0.1 | 0.3 | 0.1 | 0.1 | 0.1 | 0.1 | 0.1 |
| | TimeXer | 0.5 | 0.5 | 0.1 | 0.1 | 0.1 | 0.1 | 0.1 | 0.1 | 0.1 | 0.1 | 0.1 | 0.1 | 0.1 | 0.1 | 0.1 | 0.1 | 0.3 | 0.1 |
| | TimeBridge | 0.5 | 0.7 | 0.3 | 0.3 | 0.1 | 0.1 | 0.1 | 0.1 | 0.1 | 0.1 | 0.1 | 0.3 | 0.1 | 0.1 | 0.1 | 0.3 | 0.1 | 0.1 |
| ETTm2 | TimeMixer | 0.1 | 0.3 | 0.1 | 0.1 | 0.1 | 0.1 | 0.1 | 0.1 | 0.1 | 0.1 | 0.1 | 0.1 | 0.1 | 0.3 | 0.1 | 0.3 | 0.1 | 0.1 |
| | TimesNet | 0.1 | 0.1 | 0.3 | 0.1 | 0.1 | 0.1 | 0.1 | 0.1 | 0.1 | 0.1 | 0.1 | 0.1 | 0.1 | 0.1 | 0.1 | 0.1 | 0.1 | 0.1 |
| | TimeXer | 0.3 | 0.3 | 0.5 | 0.3 | 0.1 | 0.1 | 0.1 | 0.1 | 0.1 | 0.1 | 0.1 | 0.1 | 0.1 | 0.1 | 0.1 | 0.1 | 0.1 | 0.3 |
| | TimeBridge | 0.1 | 0.1 | 0.1 | 0.1 | 0.1 | 0.1 | 0.1 | 0.3 | 0.1 | 0.1 | 0.1 | 0.1 | 0.1 | 0.1 | 0.1 | 0.1 | 0.1 | 0.3 |

Table 12: Found Correction Strength $\beta$ for UEC Models Across Datasets and Backbones (cont.)

| Dataset | Backbone | STD (Ours) | | MLP | | Logistic | | RF | | XGB | | LSTM | | GRU | | CNN | | TF. | |
|---|---|---|---|---|---|---|---|---|---|---|---|---|---|---|---|---|---|---|---|
| | | MSE | MAE | MSE | MAE | MSE | MAE | MSE | MAE | MSE | MAE | MSE | MAE | MSE | MAE | MSE | MAE | MSE | MAE |
| Traffic | TimeMixer | 0.1 | 0.1 | 0.1 | 0.1 | N/A | N/A | N/A | N/A | N/A | N/A | 0.1 | 0.1 | 0.1 | 0.1 | 0.1 | 0.3 | 0.3 | 0.1 |
| | TimesNet | 0.3 | 0.1 | 0.1 | 0.1 | N/A | N/A | N/A | N/A | N/A | N/A | 0.1 | 0.1 | 0.1 | 0.1 | 0.1 | 0.1 | 0.1 | 0.1 |
| | TimeXer | 0.1 | 0.5 | 0.1 | 0.1 | N/A | N/A | N/A | N/A | N/A | N/A | 0.1 | 0.1 | 0.1 | 0.1 | 0.1 | 1.0 | 0.1 | 1.0 |
| | TimeBridge | 0.1 | 0.1 | 0.1 | 0.1 | 0.1 | 0.1 | 0.1 | 0.3 | 0.1 | 0.1 | 0.1 | 0.1 | 0.1 | 0.1 | 0.1 | 0.1 | 0.1 | 0.3 |
| Weather | TimeMixer | 0.1 | 0.1 | 0.1 | 0.1 | 0.1 | 0.1 | 0.1 | 0.1 | 0.1 | 0.1 | 0.3 | 0.1 | 0.3 | 0.3 | 0.1 | 0.1 | 0.3 | 0.1 |
| | TimesNet | 0.1 | 0.1 | 0.1 | 0.1 | 0.1 | 0.1 | 0.5 | 0.1 | 0.3 | 0.1 | 0.5 | 0.3 | 0.5 | 0.3 | 0.1 | 0.1 | 0.3 | 0.3 |
| | TimeXer | 0.1 | 0.1 | 0.1 | 0.1 | 0.1 | 0.1 | 0.1 | 0.1 | 0.1 | 0.1 | 0.1 | 0.1 | 0.1 | 0.1 | 0.1 | 0.1 | 0.1 | 0.1 |
| | TimeBridge | 0.1 | 0.1 | 0.1 | 0.1 | 0.1 | 0.1 | 0.1 | 0.1 | 0.1 | 0.1 | 0.1 | 0.1 | 0.1 | 0.1 | 0.1 | 0.1 | 0.1 | 0.1 |
| Electricity | TimeMixer | 0.1 | 0.1 | 0.3 | 0.1 | N/A | N/A | N/A | N/A | N/A | N/A | 0.1 | 0.1 | 0.1 | 0.1 | 0.1 | 0.1 | 0.1 | 0.1 |
| | TimesNet | 0.3 | 0.1 | 0.1 | 0.1 | N/A | N/A | N/A | N/A | N/A | N/A | 0.1 | 0.1 | 0.1 | 0.1 | 0.1 | 0.1 | 0.1 | 0.1 |
| | TimeXer | 0.1 | 0.1 | 0.1 | 0.3 | N/A | N/A | N/A | N/A | N/A | N/A | 0.1 | 0.1 | 0.1 | 0.1 | 0.1 | 1.0 | 0.1 | 1.0 |
| | TimeBridge | 0.1 | 0.1 | 0.1 | 0.1 | 0.1 | 0.1 | 0.1 | 0.1 | 0.1 | 0.1 | 0.1 | 0.1 | 0.1 | 0.1 | 0.1 | 0.1 | 0.1 | 0.1 |

Table 13: Performance comparison (averaged MSE and MAE across prediction lengths 96, 192, 336, and 720) for AR and DF methods on different datasets and models.

| Dataset | Model | DF (MSE / MAE) | AR (MSE / MAE) |
|---|---|---|---|
| ETTh1 | TimeMixer | 0.4490 / 0.4399 | **0.4357 / 0.4348** |
| ETTh1 | TimesNet | 0.4879 / 0.4722 | **0.4715 / 0.4655** |
| Weather | TimeMixer | **0.2445** / 0.2748 | 0.2446 / **0.2739** |
| Weather | TimesNet | **0.2634 / 0.2910** | 0.2699 / 0.2964 |
| Traffic | TimeMixer | **0.5041 / 0.3241** | 0.5485 / 0.3385 |
| Traffic | TimesNet | 0.7606 / 0.4419 | **0.7014 / 0.3991** |

Table 14: Results on ETTh1 dataset with different training losses of UEC across backbones. Bold denotes the best results.

| Backbone | Huber | | L1 | | MSE | |
|---|---|---|---|---|---|---|
| | MSE | MAE | MSE | MAE | MSE | MAE |
| TimeMixer | **0.434** | **0.435** | **0.434** | 0.438 | **0.434** | 0.438 |
| TimesNet | **0.534** | **0.488** | 0.536 | 0.491 | 0.535 | 0.490 |

Table 15: Average MSE and MAE across all prediction lengths $\{96, 192, 336, 720\}$ for ETTm1 dataset using TimeMixer as backbone, with different kernel sizes $\{5, 25, 50\}$.

| Kernel size | MSE | MAE |
|---|---|---|
| 5 | 0.4060 | 0.4136 |
| 25 | 0.4048 | 0.4184 |
| 50 | 0.4044 | 0.4211 |

Table 16: Mean and standard deviation of MSE and MAE for UEC-STD and TimeMixer (3 runs).

| Method | MSE (mean $\pm$ std) | MAE (mean $\pm$ std) |
|---|---|---|
| UEC-STD | $0.4273 \pm 0.0029$ | $0.4343 \pm 0.0012$ |
| TimeMixer | $0.4357 \pm 0.0006$ | $0.4343 \pm 0.0006$ |

Table 17: Performance gains (MAPE, %) on ETTh1 and ETTm1.

| Method | ETTh1 | ETTm1 |
|---|---|---|
| AR (No Correction) | 0.00 | 0.00 |
| UEC-MLP | -3.61 | -0.58 |
| UEC-Logistic | -12.48 | -2.09 |
| UEC-Random Forest | 7.02 | -1.74 |
| UEC-XGBoost | -9.89 | **-4.42** |
| UEC-LSTM | -9.62 | -0.59 |
| UEC-GRU | -10.27 | -0.83 |
| UEC-CNN | -8.42 | -1.07 |
| UEC-TF | 3.04 | -1.10 |
| UEC-STD | **-21.66** | -2.16 |

Table 18: MSE results on the US_Births dataset for different models and prediction horizons.

| Horizon | AR | MLP | Logistic | Random Forest | XGBoost | LSTM | GRU | CNN | TF | STD |
|---|---|---|---|---|---|---|---|---|---|---|
| 96 | 0.2303 | 0.2265 | 0.2149 | 0.2231 | 0.2219 | 0.2187 | 0.2146 | 0.2282 | 0.2185 | 0.2008 |
| 192 | 0.2662 | 0.2598 | 0.2370 | 0.2514 | 0.2469 | 0.2462 | 0.2428 | 0.2622 | 0.2493 | 0.2273 |
| 336 | 0.2806 | 0.2732 | 0.2493 | 0.2631 | 0.2583 | 0.2585 | 0.2564 | 0.2734 | 0.2618 | 0.2309 |
| 720 | 0.3067 | 0.2965 | 0.2541 | 0.2775 | 0.2656 | 0.2759 | 0.2723 | 0.3108 | 0.2788 | 0.2405 |
| Average | 0.2709 | 0.2640 | 0.2388 | 0.2537 | 0.2481 | 0.2498 | 0.2465 | 0.2686 | 0.2521 | **0.2248** |

