# OpenReview forum: "Reviving Error Correction in Modern Deep Time-Series Forecasting"
_ICLR.cc/2026/Conference — Submitted to ICLR 2026_

### Official Review · Reviewer_uD85 · 2025-10-24

**Soundness:** 2
**Presentation:** 3
**Contribution:** 2
**Rating:** 4
**Confidence:** 4

**Summary:**

This paper revisits the issue of error accumulation in deep time-series forecasting and proposes a universal, architecture-agnostic Universal Error Corrector (UEC) and its variant UEC-STD with seasonal–trend decomposition. The method can be plugged into existing forecasting backbones without retraining and significantly reduces long-horizon prediction errors. The authors provide theoretical motivation, clear formulation, and extensive experiments across multiple datasets and backbones, showing consistent improvements in accuracy and robustness.

**Strengths:**

1. Architecture-agnostic Universal Error Corrector (UEC) can work with various deep backbones without retraining or architecture modification, and needs low training overhead.
2. The experimental results demonstrate that the proposed method achieves strong performance.
3. The paper is well-structured, with a logical flow from motivation to methodology and experiments.

**Weaknesses:**

1. The entire UEC module is trained on the validation set. This data was already used to select the best checkpoint for the backbone model. Training a new model component on this same data risks overfitting to the specific error patterns of the validation set and does not guarantee generalization to the test set.
2. The UEC-STD's success hinges on a moving-average decomposition with a fixed, hard-coded kernel size of 25. This choice is never justified, ablated, or discussed. What if the data has a different seasonality, or no seasonality at all? The method's performance is fundamentally tied to this single, arbitrary hyperparameter.
3.  The correction strength $\beta$ seems to be a critical crutch for the model. Table 8 shows that the automatically-selected $\beta$ is often very small (e.g., 0.1 or 0.3). This implies that the raw output of the UEC is often highly noisy, and its utility is only realized by severely damping its predictions.

**Questions:**

1. Can the authors provide a stronger justification for training on the validation set? How can we be confident that the UEC is not simply overfitting to the validation set's error distribution?
2. What is the performance of UEC-STD if $\beta$ is fixed to 1.0 for all experiments?
3. Please provide an ablation study on the moving average kernel size (ks=25). How sensitive is the model's performance to this parameter? What happens if ks is set to 5, or 50, or chosen based on a dataset's known seasonal period?

---

> ### Author Response · Authors · 2025-11-21
> **Response to Reviewer uD85**
>
> Thank you for your careful review. We address your concerns and questions as follows.
> - **"The entire UEC module ..."** and **"Can the authors provide ..."**. To mitigate the potential overfitting, we introduced the correction strength β. As explained in Sec. 2.2 On Choosing the Correction Strength, β is automatically selected based on a more diverse subset of data, rather than relying solely on the validation set. This ensures that UEC-STD captures broadly applicable correction patterns and can generalize better, as reflected by consistent improvements across all test datasets.
> - **"The UEC-STD's success ..."** and **"Please provide an ablation ..."**. hank you for raising this point. We follow prior works [1] to set the kernel size as 25 to save hyperparameter tuning cost. In this revision (appendix Table 15), we have added an ablation study on ETTm1 and TimeMixer to examine the impact of the hyperparameter. Overall, the results suggest that the optimal kernel size can vary depending on the evaluation metric, MSE or MAE, and 25 is a reasonable value.  Setting this hyperparameter to 25 can reduce tuning costs, and further tuning could potentially further improve the performance of our method.
>  | Kernel size | MSE     | MAE     |
> |--------|---------|---------|
> | 5      | 0.4060  | 0.4136|
> | 25     | 0.4048  | 0.4184 |
> | 50 | 0.4044  | 0.4211 |
> - **"The correction strength ..."** and **"What is the performance of ..."**. The fact that the automatically selected β is often small and differs from the value of 1 used during training reflects a mismatch between the training and testing environments. Essentially, the model tends to overcorrect when applying patterns learned from the training data. We emphasize that the raw output of UEC is not noisy. It still captures the correct direction for error correction, but using a larger magnitude can lead to overcompensation. This underscores the importance of our automatic β selection mechanism, which mitigates overcorrection and identifies a reasonable correction strength that generalizes well to unseen data.
> If β is fixed to 1.0 for all experiments, the performance of UEC-STD will degrade in some datasets. As explained above, using a fixed β ignores the mismatch between training and testing distributions, leading the corrector to overcompensate for errors observed during training. This results in overcorrection, which is why our automatic β selection mechanism is critical for robust performance.
>
> [1] Zeng, Ailing, Muxi Chen, Lei Zhang, and Qiang Xu. "Are transformers effective for time series forecasting?." In Proceedings of the AAAI conference on artificial intelligence, vol. 37, no. 9, pp. 11121-11128. 2023.

---

> > ### Comment · Reviewer_uD85 · 2025-11-26
> >
> > I thank the authors for their thoughtful and detailed rebuttal. I appreciate the effort put into clarifying the experimental setup and the motivation behind the method. However, after careful consideration, I remain unconvinced regarding the overall novelty and technical contribution of the proposed approach. Therefore, I will be keeping my initial score.

---

### Official Review · Reviewer_BJsk · 2025-10-31

**Soundness:** 2
**Presentation:** 3
**Contribution:** 2
**Rating:** 4
**Confidence:** 4

**Summary:**

The proposed UEC-STD framework is novel, practical, and demonstrates consistent improvements across diverse settings. While some theoretical aspects could be strengthened and a broader evaluation would be beneficial, the work provides both immediate practical utility and valuable insights for future research.
The architecture-agnostic nature of the approach is particularly valuable, as it enables practitioners to improve existing models without the computational cost of retraining. The seasonal-trend decomposition strategy is well-motivated and effectively implemented.

**Strengths:**

1. Architecture-Agnostic Design: The proposed Universal Error Corrector (UEC) framework can be integrated with any existing forecaster without retraining, addressing a major limitation of previous approaches
2. Seasonal-Trend Decomposition (UEC-STD): The explicit separation of trend and seasonal components for targeted correction is theoretically sound and aligns with established time series analysis principles
3. Practical Implementation: The method requires minimal computational overhead (∼10% of backbone training time) while delivering consistent improvements

**Weaknesses:**

1. Methodological Limitations:
- Decomposition Simplicity: The moving average approach for seasonal-trend decomposition, while practical, may be too simplistic for complex, non-stationary time series
- Hyperparameter Sensitivity: The method requires careful tuning of λs-λt coefficients and correction strength β, which could limit out-of-the-box applicability
- Assumption of Decomposability: The approach assumes that time series can be cleanly separated into trend and seasonal components, which may not hold for all real-world data
2. Experimental Considerations:
- Limited Real-World Testing: While benchmark datasets are comprehensive, additional testing on more diverse, noisy real-world datasets would strengthen the claims
- Computational Overhead: Although minimal compared to backbone training, the additional inference time (UEC prediction at each autoregressive step) could be problematic for real-time applications
- Scalability Concerns: The method's performance on very high-dimensional multivariate time series needs further validation

**Questions:**

see weaknesses

---

> ### Author Response · Authors · 2025-11-21
> **Response to Reviewer BJsk**
>
> We thank the reviewer for their valuable feedback. We address your concerns and questions point by point below.
> - **"Decomposition Simplicity:  ..."**. While the moving average approach for seasonal-trend decomposition is relatively simple, it is a good choice to start with as pointed out by previous studies [1]. In practice, our experiments show that it is highly effective. It reliably separates trend and seasonal components for a wide range of datasets, enabling the corrector to focus on meaningful residual errors without introducing unnecessary complexity. Moreover, moving-average-based decomposition is widely used in modern time-series forecasting for diverse datasets [2,3], highlighting its practicality and robustness even for non-stationary time series.
> - **"Hyperparameter Sensitivity: ..."**. We note that β is automatically selected, not tuned (see Sec. 2.2 On Choosing the Correction Strength). Furthermore, as analyzed in Sec. 4.2, our suggested starting point of λ_s = λ_t = 0.5 provides a robust guideline that works well in practice and can be further adjusted if needed, making the method reasonably straightforward to apply.
> - **"Assumption of Decomposability: "**. While our method assumes that the time series can be decomposed into trend and seasonal components (which is a common assumption in the literature), the decomposition acts as a practical inductive bias that helps the corrector focus on systematic patterns, rather than requiring a perfect separation. Our improved results on real-world data like electricity and weather validate the practicality of our approach.
> - **"Limited Real-World Testing: ..."**. Thanks for your suggestions. We have added additional experiments on the US Births dataset (https://zenodo.org/records/4656049). As reported in Appendix Table 18, our method UEC-STD significantly outperforms the backbone TimeMixer and other UEC-based approaches, confirming the effectiveness of UEC-STD. Below, we summarize the performance gain in %:
> | Method             | US_Births |
> |-------------------|-----------|
> | AR                 | 0         |
> | UEC-MLP            | -2.49     |
> | UEC-Logistic       | -11.48    |
> | UEC-Random Forest  | -6.11     |
> | UEC-XGBoost        | -8.05     |
> | UEC-LSTM           | -7.61     |
> | UEC-GRU            | -8.85     |
> | UEC-CNN            | -0.47     |
> | UEC-TF             | -6.81     |
> | UEC-STD            | **-16.67**    |
> -  **"Computational Overhead: ..."**. In practice, the extra computation adds only a small fraction of the total runtime, making it suitable for many near-real-time scenarios. For strict latency-constrained applications, the corrector could also be parallelized or applied in a batched fashion, further mitigating the overhead.
> - **"Scalability Concerns: ..."**. n our experiments, we evaluated UEC-STD on high-dimensional datasets, such as Traffic, which contain hundreds of features. The method consistently improves performance in these settings, indicating that UEC-STD can handle realistic multivariate scenarios. Notably, UEC-STD scales effectively, whereas more naive approaches, such as XGBoost, struggle to scale in these high-dimensional cases.
>
> [1] Kreuzer, T., Zdravkovic, J. & Papapetrou, P. Unpacking the trend: decomposition as a catalyst to enhance time series forecasting models. Data Min Knowl Disc 39, 54 (2025). https://doi.org/10.1007/s10618-025-01120-8
>
> [2] Zeng, Ailing, Muxi Chen, Lei Zhang, and Qiang Xu. "Are transformers effective for time series forecasting?." In Proceedings of the AAAI conference on artificial intelligence, vol. 37, no. 9, pp. 11121-11128. 2023.
>
> [3]  Wu, Haixu, Jiehui Xu, Jianmin Wang, and Mingsheng Long. "Autoformer: Decomposition transformers with auto-correlation for long-term series forecasting." Advances in neural information processing systems 34 (2021): 22419-22430.

---

### Official Review · Reviewer_5psk · 2025-11-01

**Soundness:** 2
**Presentation:** 2
**Contribution:** 1
**Rating:** 2
**Confidence:** 4

**Summary:**

This paper addresses error accumulation in deep learning-based time-series forecasting during autoregressive inference, where predictions are recursively used as inputs. The authors propose the Universal Error Corrector with Seasonal-Trend Decomposition (UEC-STD), a simple and architecture-agnostic post-processing module that can enhance any pre-trained forecasting model without requiring retraining. UEC-STD explicitly decomposes predictions into trend and seasonal components and learns separate corrections for each, optimizing a weighted loss function. The method is trained on validation data while keeping the backbone forecaster frozen, making it practical and efficient. Experiments across 7 datasets and 3 modern forecasting models (TimeMixer, TimesNet, TimeXer) demonstrate that UEC-STD consistently reduces both MSE and MAE by approximately 2.1% and 0.8% on average, with particularly strong improvements on datasets like ETTm1 (4.78% MSE reduction), while adding minimal computational overhead (roughly 10% of backbone training time).

**Strengths:**

1. Error correction models are indeed an important aspect of forecasting that is currently missing from deep-learning-based approaches2. 2. The ablation study is well-designed and comprehensive

**Weaknesses:**

1. The presentation of Figure 2 is unclear, making it difficult for the reader to interpret the key takeaways.
2. Averaging results across all backbones in Tables 1 and 3 may obscure performance nuances. (No one will use all backbones at the same time in practice) A more informative presentation would be to disaggregate these results, showing performance for each backbone individually. Additionally, tracking the number of "first place" finishes per backbone would offer a clearer comparison of methods.
3. The reported improvements are modest, making it difficult to assess their significance. To validate these gains, please provide statistical significance tests. Furthermore, the 2% gain is not a lot when compared to 14% error increase depicted in Figure 1(b).
4. An inconsistency arises as only the UEC-STD variant shows consistent improvement, while other related UEC architectures do not. This selective improvement is counter-intuitive and requires further analysis or explanation.

**Questions:**

1. What are the look-back length, per-step forecast horizon, and number of steps in the main experiments?
2. The analysis could be strengthened by examining performance trends over time. It would be valuable to see if the method's improvement (relative to baselines) changes as the forecast horizon extends deeper into the future.

---

> ### Author Response · Authors · 2025-11-21
> **Response to Reviewer 5psk**
>
> Thank you for your thoughtful review. We address your concerns and questions point by point below.
> - **"The presentation of Figure 2 ..."**. We believe Figure 2 already clearly conveys the core message: the corrector refines a pre-trained forecaster by decomposing both the forecast and its error into trend and seasonal components and applying component-wise corrections. Subfigures (a), (b), and (c) respectively show the overall workflow, the architecture, and the supervised training process. We have nonetheless improved the caption to make this flow even easier to interpret in the revised manuscript.
> - **"Averaging results across ..."**. Due to space constraints, we reported the raw score for each backbone in Appendix Tables 4,5, and 6 a tracking the number of “first place” as well as “second place”.
> - **"The reported improvements are ..."**. We note that the 14% figure represents a theoretical upper bound on the performance a corrector could achieve, and therefore is not a practical baseline for comparison. The gap between this theoretical limit and actual performance highlights that error correction remains an open problem requiring further investigation. As explained, the observed 2% gain is already meaningful, as it is comparable to the improvements achieved by recent standalone SOTA models over prior baselines [1]. Previous studies also indicate that variance in time-series forecasting tasks is low [1], so it is unlikely to affect the significance of our improvement. To further confirm the significance of our results, and following your recommendation,  in Appendix Table 16, we have added experiments with mean and std. (3 runs) of UEC+STD (TimeMixer) against TimeMixer for ETTh1 and ETTm1. As we can see, the variance is very small, and UEC-STD's MSE  is significantly better than TimeMixer's with a p-value of 0.0165 and 0.0071 for ETTh1 and ETTm1, respectively:
> | Metric | Dataset | UEC-STD (mean ± std) | TimeMixer (mean ± std) |
> |--------|---------|--------------------|-----------------------|
> | MSE | ETTh1   | 0.4273 ± 0.0029    | 0.4357 ± 0.0006       |
> | MSE | ETTm1   | 0.4153 ± 0.00094   | 0.4217 ± 0.00189      |
>
> - **"An inconsistency arises ..."**. We do not view the selective improvement as counter-intuitive; rather, it reflects the inherent difficulty of the error-correction problem. Simply attaching an arbitrary architecture as a corrector does not guarantee gains. This underscores the importance of carefully designed structures like UEC-STD, which explicitly leverage the unique characteristics of time-series data (trend and seasonal), in providing consistent, reliable corrections to backbone forecasters.
> - **"What are the look-back ..."**. We mentioned that in Fig. 1’s caption and the experiment sections.
>   + look-back length: W=96
>   + per-step forecast horizon: L=96, which is the fixed prediction window used by the backbone during autoregressive rollout.
>   + number of steps: 96, 192, 336, 720, representing the total number of future points to be predicted.
> We report the averaged results across these horizons in Table 1.
> - **"The analysis could be ..."**. Thank you for your suggestion. The raw numbers in Appendix Tables 4, 5, and 6 indicate that improvements generally tend to grow as the forecast horizon increases, as expected, because short horizons experience far less autoregressive error accumulation. For example, we visualize the relative performance gain with the TimeMixer backbone on ETTh1 and ETTm1 in Appendix Fig. 6 in the revised manuscript. Here, for ETTm1, gains are always larger for longer forecast horizons (with the shortest horizon 96 where no autoregression is used, correction even degrades performance), while for ETTh1, the improvement all increases and peaks at  L=336.
>
> [1] Wang, Yuxuan, Haixu Wu, Jiaxiang Dong, Guo Qin, Haoran Zhang, Yong Liu, Yunzhong Qiu, Jianmin Wang, and Mingsheng Long. "Timexer: Empowering transformers for time series forecasting with exogenous variables." Advances in Neural Information Processing Systems 37 (2024): 469-498.

---

### Official Review · Reviewer_FFS7 · 2025-11-01

**Soundness:** 3
**Presentation:** 3
**Contribution:** 2
**Rating:** 4
**Confidence:** 2

**Summary:**

The authors propose a method named Universal Error Corrector with Seasonal-Trend Decomposition (UEC-STD), which leverages error correction mechanisms for time series forecasting and can be used to enhance model performance. The authors apply their proposed method to baseline models and show that it improves the performance in prediction accuracy.

**Strengths:**

1. Overall this paper is clearly written and easy to understand.

2. The proposed method using error correction mechanism is moderately novel and may be potentially applicable to similar problems in this domain.

**Weaknesses:**

1. The baseline models in Table 1 do not represent the state-of-the-art (SOTA) methods for time series forecasting. In order to justify the significance of this research contribution, the authors should clearly demonstrate that the proposed UEC framework can further improve a wide range of SOTA methods.

2. In addition to MAE and MSE, the authors should evaluate their proposed method with MAPE (mean absolute percentage error) which is robust under different scales of the time series values.

3. The authors should also evaluate their proposed method on standard benchmark datasets for time series forecasting, such as the M4 competition dataset.

4. Recent works in time series forecasting should be reviewed in Related Works.

**Questions:**

1. Why are some evaluation results not available in Table 1?

2. How does UEC perform when it is applied to the SOTA methods for deep time series forecasting?

---

> ### Author Response · Authors · 2025-11-21
> **Response to Reviewer FFS7**
>
> Thank you for your constructive review. We address your concerns and questions point by point below.
> - **"The baseline models in Table 1 ..."**. Our baseline forecasters were selected because they are lightweight, well-established, and suitable for our hardware while delivering strong performance. They are also standardized and widely implemented in major open-source forecasting libraries such as TSLib, making them far more representative of real-world practice. While more complex models may offer performance gains, they are not necessarily reproducible or widely adopted in practice.  We also note that our models were not extensively tuned. For instance, we simply used the default history window of 96 across all backbones and datasets, which can make the results appear a bit weaker compared to heavily tuned SOTA models. That said, to address your concern, we have also applied our method to TimeBridge [1], a more recent model in the revision. The results (see Appendix Tables 7, 8, and 9) show that our method is still able to improve the recent model effectively. Here is the summary:
> | Dataset / Metric | MSE Improvement % | MAE Improvement %|
> |-----------------|----------------|----------------|
> | ETTh1           | -1.15          | 0.13           |
> | ETTh2           | -1.03          | -0.61          |
> | ETTm1           | -1.92          | -0.92          |
> | ETTm2           | -7.63          | -4.82          |
> | Traffic         | -1.91          | -2.13          |
> | Weather         | -1.43          | -0.40          |
> | Electricity     | 0.02           | -0.08          |
> | **Average**     | **-2.58**      | **-1.14**      |
>
> - **"In addition to MAE and MSE ..."**. Thanks for your suggestions. We have incorporated MAPE in the revision (see Appendix Table 17 for representative ETTh1 and ETTm1 datasets). The results show that UEC-STD consistently achieves the strongest overall improvements (top 1 and 2), outperforming all other correction strategies by a clear margin. Notably, when evaluated under the MAPE metric, the relative gains of UEC-STD can become even more pronounced, indicating that its advantages are larger on scale-robust error measures.
> | Method          | ETTh1 (MAPE %)| ETTm1 (MAPE %) |
> |----------------------|--------------------|---------------------|
> | AR (No correction)                  | 0                  | 0                   |
> | UEC-MLP              | -3.61              | -0.58               |
> | UEC-Logistic         | _-12.48_           | -2.09               |
> | UEC-Random Forest    | 7.02               | -1.74               |
> | UEC-XGBoost          | -9.89              | **-4.42**           |
> | UEC-LSTM             | -9.62              | -0.59               |
> | UEC-GRU              | -10.27             | -0.83               |
> | UEC-CNN              | -8.42              | -1.07               |
> | UEC-TF               | 3.04               | -1.10               |
> | **UEC-STD**          | **-21.66**         | _-2.16_             |
> - **"The authors should also evaluate ..."**. The M4 competition dataset is more suitable for short-term forecasting, which does not reveal the limitations of autoregressive inference and falls outside our focus on long-term forecasting. Instead, we have added new experiments with the US Births Dataset (https://zenodo.org/records/4656049)
> in this revision. As reported in Appendix Table 18, our method UEC-STD significantly outperforms the backbone TimeMixer and other UEC-based approaches, confirming the effectiveness of UEC-STD. Below, we summarize the performance gain in %:
> | Method             | US_Births |
> |-------------------|-----------|
> | AR  (No correction)               | 0         |
> | UEC-MLP            | -2.49     |
> | UEC-Logistic       | -11.48    |
> | UEC-Random Forest  | -6.11     |
> | UEC-XGBoost        | -8.05     |
> | UEC-LSTM           | -7.61     |
> | UEC-GRU            | -8.85     |
> | UEC-CNN            | -0.47     |
> | UEC-TF             | -6.81     |
> | UEC-STD            | **-16.67**    |
> - **"Recent works in ..."**. Our paper primarily targets error-correction techniques, so the related-work discussion naturally centers on prior work in this area. Regarding the backbone forecasters, we included several representative and well-established works. We have added additional references in the revision and hope this addresses your concern.
> - **"Why are some evaluation ..."**. As explained in the caption of the Table 1, N/A indicates that the method failed to converge or crashed during training.
> - **"How does UEC ..."**. As shown in above, UEC still helps improve SOTA methods like TimeBridge by 2.58% and 1.14% in MSE and MAE, respectively, which is significant in the field.
>
> [1] [1] Peiyuan Liu, Beiliang Wu, Yifan Hu, Naiqi Li, Tao Dai, Jigang Bao, and Shu-Tao Xia. Timebridge:
> Non-stationarity matters for long-term time series forecasting. International Conference on Machine Learning, 2025

---

### Author Response · Authors · 2025-11-21
**General Response to Reviewers**

We greatly appreciate the reviewers' thorough assessment of our paper. We are very encouraged that the reviewers found our contribution to be significant, specifically praising the novelty of our method (FFS7), the clarity of our presentation (FFS7, uD85), and the fact that we are addressing an important problem (5psk). Furthermore, we are pleased that our approach was recognized as sound, efficient, and versatile across diverse backbones (BJsk, uD85), and that our experimental results were deemed strong and comprehensive (5psk, uD85).

We have carefully addressed all remaining concerns in the individual responses below. We respectfully ask the reviewers to consider raising their scores if our response is valid.

Below we summarize the change in the revision:
- New experiments on new backbone (TimeBridge) and dataset (US Briths)
- Report additional metric MAPE
- Report results with statistics over multiple runs
- Ablation study on kernel size of moving average
- Improve clarity of writing

---

### Meta-Review · Area_Chair_DwuP · 2025-12-28

**Summary:**

The paper proposed an error correction mechanism for deep time-series forecasting models called Universal Error Corrector with Seasonal–Trend Decomposition (UEC-STD).

All reviewers have concern on the weak experiments. Reviewer **FFS7** noted that the baseline methods are not state-of-the-art and the author also did not conduct experiment on standard benchmarks like M4. This aligned with Reviewer **5psk**’s review about modest improvement and Reviewer **BJsk**’s concern on limited real-world testing. Reviewer **uD85** has concern that the UEC module is trained on the validation set and is prune to overfitting and also has concern on the novelty of the method.

Given the unaddressed concerns, the paper is clearly behind the bar of ICLR so I’m voting for rejection.

**Reviewer Concerns:**

For reviewer **FFS7**, the question about method’s performance in MAPE has been addressed with the new experiments. Other concerns are still outstanding.

For reviewer **5psk**, the concerns on experiments are still outstanding.

For reviewer **BJsk**, the concerns around methodology limitations and experiments are outstanding.

For reviewer **uD85**, the concern on hard-coded kernel size has been addressed by the new experiments. The other concerns are outstanding.

**Reviewer Scores:**

Reviewers may keep their scores.

---

### Decision · Program_Chairs · 2026-01-26

Reject